# DEEP SYNCHRONISATION-BASED CLUSTERING

## ABSTRACT

Identifying patterns in high-dimensional and complex data, such as images, requires techniques that extract meaningful features. Deep clustering combines the representation power of neural networks with classical clustering and has shown strong performance on such data. However, most approaches build on $k$-Means, inheriting its assumptions about cluster shapes, requiring the number of clusters in advance, and lacking an intuitive stopping criterion. We propose DeepSynC, the first synchronisation-based deep clustering algorithm that overcomes these limitations. It begins by identifying core points in the embedded space and assigning them to clusters. A novel cluster loss then synchronises similarly embedded objects, enabling the gradual assignment of further points. This combination of synchronisation-based loss and assignment strategy allows greater flexibility in cluster shape and introduces an automatic stopping condition for training.

## 1 INTRODUCTION

Identifying patterns in large, high-dimensional data structures, such as videos and images, is challenging. Even more so in the absence of domain knowledge or labels. Clustering is an unsupervised task that aims at grouping similar data objects while separating dissimilar ones. Classical methods such as $k$-Means (Lloyd, 1982) or DBSCAN (Ester et al., 1996) suffer from the curse of dimensionality (Indyk & Motwani, 1998) when applied to such high-dimensional data. Thus, analysing such data requires elaborate techniques to automatically extract a subset of appropriate features.

In the past decade, the representation learning capabilities of neural networks have been exploited for this task and have led to a research field termed *Deep Clustering*. By extracting a lower-dimensional representation of the data, they become feasible for subsequent analysis with classical clustering approaches. E.g., SCDE (Duan et al., 2019) applies spectral clustering after embedding the data, and SHADE (Beer et al., 2024) learns a representation tailored to a subsequent density-connectivity clustering approach. However, simultaneous optimisation of feature learning and clustering outperforms these sequential methods. This has been shown, for instance, in Xie et al. (2016) for $k$-Means clustering as intermediate cluster labels help to shape the embedding. Often, these simultaneous approaches use a reconstruction-based representation learning module in the form of an autoencoder (AE). The learning protocol usually requires pretraining the AE and clustering in the embedded space to obtain initial labels for the objects. Subsequently, the assignments and the embedding are updated to obtain a 'cluster-friendly' embedding by adding a clustering objective $\mathcal{L}_{clu}$ to the reconstruction loss. The majority of algorithms in this family focuses on a simple centroid-based cluster objective. Although $k$-Means-like clustering has several benefits the cluster assumptions are quite strong: Clusters are assumed to be spherical, and the number of clusters needs to be specified in advance. However, in modern high-dimensional data, clusters are often of more complex shape. To overcome these limitations, we propose to incorporate a different concept into deep clustering: Synchronisation. Synchronisation occurs naturally in many fields of research, such as neuroscience (Bauer et al., 2022) or chemistry (Liu et al., 2022). It refers to the observed phenomenon that individual units align their intrinsic behaviours after mutual interaction. Several models have been proposed to mathematically describe synchronisation, of which the Kuramoto model (Kuramoto, 1975; Acebrón et al., 2005) has proven suitable to adapt it to clustering (e.g. Shao et al. (2017a;b)). The interacting units are modelled as phase oscillators that can be thought of as vectors moving counter-clockwise around the unit circle. Their momentary phase is their current position on the unit circle. Every oscillator has its own intrinsic frequency of movement. The Kuramoto model then describes the temporal changes in their phase values, once they interact with each other. If the

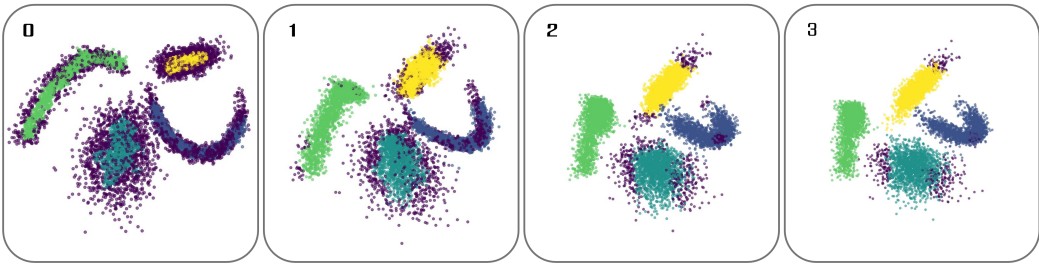

Figure 1: 2D PCA projections of DeepSynC's 3D latent space at different epochs for a 3D toy dataset with 4 ground-truth clusters of varying shapes, distributions, and border densities. DeepSynC correctly clusters the data, gradually assigns points, and leaves ambiguous ones unassigned (purple).

interaction is strong, they start to align their movement. This model has been exploited for the first time in the context of clustering by Böhm et al. (2010). Each data object is regarded as a phase oscillator, and every coordinate of each data point changes following an iteration rule derived from the Kuramoto model.

An important measure derived from the Kuramoto model is the Kuramoto order parameter (KOP) $r \in [0, 1]$, which measures the strength of the (local) synchronisation between data points. In the context of clustering, $r = 1$ implies that all points within a cluster have the same position.

Moving points based on synchronisation with neighbours and clustering once local synchronisation is reached has proven effective in many classic data mining tasks like co-clustering and subspace clustering (Shao et al., 2017a;b) or clustering multivariate data (Crnkić & Jaćimović, 2019). However, transferring the Kuramoto model to deep learning is challenging and, thus, it has not yet been exploited for deep clustering. In this paper, we incorporate synchronisation into a novel deep clustering algorithm by developing a new cluster loss formulation inspired by the guiding measure for synchronisation, the KOP. Hence, we do not require an explicit formulation of how the points should move. Their positions are learned by the neural network to form locally synchronised clusters.

Another common problem of existing deep clustering methods, irrespective of the used cluster model, is that they rely heavily on the initial clustering, which in turn relies heavily on the initial embedding: All data objects are assigned to a cluster right after pre-training and from then on guide the representation learning. Outliers or ambiguous points influence the model to the same degree as representative data objects. Here, we propose an assignment strategy that accounts for the certainty of a point's cluster membership. We assume that (locally) dense data points cover the core regions of each cluster. Hence, they are labelled with high confidence after pre-training. During our synchronisation-based training of the embedded space, points adapt their position according to their environment and get gradually assigned once they have moved closer to the core regions. This allows more flexibility regarding the cluster shape as we do not rely on centroids or a spherical cluster model. Furthermore, since the intermediate labels guide the embedding, it is beneficial to have high-confidence labelled points as the dominant factor in shaping the embedding (cf. Figure 1).

An additional benefit of this approach is that we can easily employ an automatic stopping criterion: Once either all points get assigned with high confidence or no additional points get labelled, DeepSynC stops. In summary, our main contributions are:

1. We introduce DeepSynC, the first deep clustering algorithm based on synchronisation.

2. DeepSynC first labels data points that can be assigned to a cluster with high confidence before the labels spread to points that are harder to cluster. The number of clusters $k$ is detected automatically.

3. Our synchronisation-based objective function, together with the gradual assignment strategy, enables an automatic stopping criterion. Hence, the crucial hyperparameter of the number of training epochs does not need to be specified.

4. Due to our stopping criterion, DeepSynC automatically identifies ambiguous points, leaving them unassigned at the end of training rather than mislabeling them.

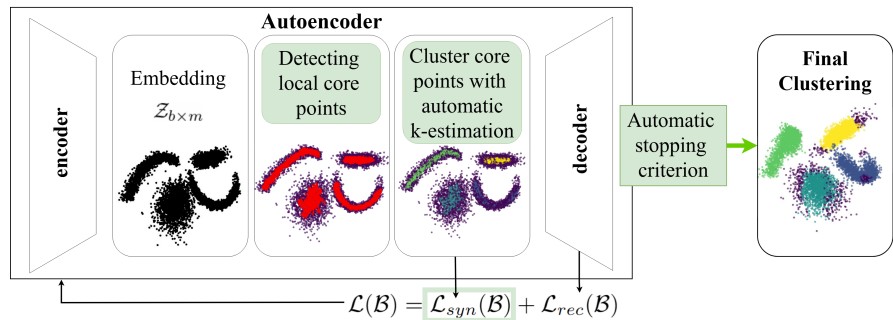

Figure 2: Overview of DeepSynC, main contributions in green.

## 2 RELATED WORK

**Interaction-based Clustering.** Classical clustering algorithms are typically categorised based on the cluster model. In this paper, we focus on models using attracting forces between objects. We do not consider centroid-based methods like $k$-Means. While DeepSynC adapts concepts of density-based methods to identify local core points, our cluster loss is solely interaction-based. Three classical clustering algorithms display a similar idea, where points adjust their positions due to attracting forces of other points: MeanShift (Comaniciu & Meer, 2002) iteratively shifts data points toward the nearest peak in the data density. Affinity Propagation (Frey & Dueck, 2007) bases point-to-point interaction on the idea of 'message passing' to decide on exemplar points. Finally, SynC (Böhm et al., 2010) moves points according to a rule derived from the Kuramoto model.

**Deep Clustering.** We focus on deep clustering methods, which typically optimise a combination of the reconstruction loss $\mathcal{L}_{rec}$ and the cluster loss $\mathcal{L}_{clu}$ that is the determining factor for the type of clusters that can be found. Most deep clustering methods like DEC (Xie et al., 2016), IDEC (Guo et al., 2017), DCN (Yang et al., 2017), DKM (Fard et al., 2020) and ACe/DeC (Miklautz et al., 2021) are based on $k$-Means, inheriting vanilla $k$-Means' restrictions regarding the shape of the clusters: Non-convex clusters with complex shapes cannot be found. Recent methods like SHADE (Beer et al., 2024), DDC (Ren et al., 2020), or DipDECK (Leiber et al., 2021) tackle this problem. However, DDC and SHADE are sequential and do not use the clustering information for the embedding.

While a large body of research focuses on optimising neural network architectures for learning representations of high-dimensional data, this is orthogonal to our approach. We regard the design of a novel clustering loss and procedure. To ensure a fair and non-overoptimised comparison across generally applicable methods, we use a basic AE for all methods. That said, the AE can be replaced by more powerful modules, e.g., contrastive learning like SimCLR (Chen et al., 2020) for specific use cases. Miklautz et al. (2025) and Leiber et al. (2025) showed that this boosts image clustering performance in methods like DEC (Xie et al., 2016) and IDEC (Guo et al., 2017). Thus, AEs remain a suitable backbone for developing and isolating the effects of new clustering objectives.

## 3 OUR NEW METHOD DEEPSYNC

As common for deep clustering, DeepSynC combines representation learning using a deep neural network with a clustering objective to learn cluster-friendly embeddings. Figure 2 illustrates the main ideas of DeepSynC: We use an autoencoder (AE) (Lecun, 1987) to learn a lower-dimensional representation of the data. This embedding is improved with the help of a novel cluster loss $\mathcal{L}_{syn}$ based on the general concept of synchronisation.

An AE is a two-part neural network, passing the input $x$ through the encoder and the decoder. The output is the reconstruction $\hat{x} = dec(enc(x))$. The goal is the embedding $z$, with a dimensionality $m$ that is much smaller than the original dimensionality $d$. The AE aims at minimising the difference between reconstruction $\hat{x}$ and input $x$. As common, DeepSynC does this in a mini-batch fashion by minimising the reconstruction error given by the mean squared Euclidean distance over the objects in a mini-batch $\mathcal{B}$: $\mathcal{L}_{rec}(\mathcal{B}) = \frac{1}{|\mathcal{B}|} \sum_{x \in \mathcal{B}} ||x - \hat{x}||_2^2$. For a notation overview, see Table 3.

## 3.1 LABEL INITIALISATION

After the network is pre-trained with $\mathcal{L}_{rec}$, we need initial labels to guide the subsequent training. While previous approaches assign all points right after pre-training, we first only assign points that exhibit a high local density, so-called *local core points*. We define them in the next paragraph.

**Local Core Points.** Some points are easier to cluster than others. They are usually at the core of the cluster, indicated by a higher density than other objects. We define a point $p$ as core if it is in a denser area than its neighbours. We can compute this regarding the point's kNN distance, also known as core distance $\kappa_p$, which is given by the distance to its $k_{neigh}$-th nearest neighbour. Core points are hard to set globally as they depend on the density of the clusters. Thus, for each point $p$, we compare its core distance $\kappa_p$ to the core distances of its $0 < T\% < 1$ nearest neighbours (denoted as $\mathcal{N}^T(p)$) – if it is smaller than the median of those, the core property is fulfilled, i.e., a point $p$ is core if and only if $\kappa_p \leq \text{med}_{q \in \mathcal{N}^T(p)} \kappa_q$. In Section A.3 (cf. Figure 7) and with ablation experiments (cf. Figure 4), we motivate the parameter $T\%$ and show the robustness of both $T\%$ and $k_{neigh}$.

**Core Point Clustering.** We now require labels for the determined core points to proceed with our simultaneous representation learning and clustering procedure. Most previous deep clustering algorithms use $k$-Means as an initial clustering. For that, one needs to specify the number of clusters $k$, and its cluster model does not align with our density-based concept for determining the core points. Therefore, we used one variant out of the SHiP-framework (Draganov et al., 2025), which captures density-connectivity features of the data (Beer et al., 2023). We choose their option of the $k$-means hierarchy and the ElbowThreshold partitioning method. This only needs to compute a similarity tree once and returns the clusterings for all possible $k$ at once. Furthermore, they have an automatic Elbow detection method which then already returns the corresponding clustering for the "best" $k$. No visual inspection of the user is required. That is why their method is very fast and performs very well. With the choice of SHiP with this setting as the algorithm to provide us with the initial cluster labels for the core points, no additional hyperparameters are introduced to our DeepSynC algorithm. Note, that the cluster number is solely determined by SHiP and does not change any more during training the network.

Theoretically, any other clustering algorithm could be chosen for this step of clustering the core points. Given the synhcronisation-based nature of DeepSynC, a straightforward choice for the initial clustering would be SynC (Böhm et al., 2010). However, for data sets such as MNIST and FMNIST calculations on the subset of local core points aborted due to the large data size, which is why it is not a suitable choice. As core points tend to be easier to cluster, DeepSynC shows increased robustness to the choice of the clustering algorithm, as supported by our results in Table 1 (provided that they can handle large sample sizes).

## 3.2 SYNCHRONISATION LOSS

In general, synchronisation can be described as the phenomenon of several units aligning their individual rhythms after interacting with each other. It has been studied in many scientific fields such as sociology, physics, biology and many others Osipov et al. (2007). The Kuramoto model is a mathematical model describing synchronisation. It is a system of ordinary differential equations modelling how a group of phase oscillators $\varphi_i$ adjust their phases over time:

$$\frac{\partial \varphi_i(t)}{\partial t} = \frac{1}{r} \sum_{j=1}^{r} C \sin(\varphi_j(t) - \varphi_i(t)) \quad i = 1, \dots, r. \tag{1}$$

Each oscillator has its own natural rhythm but interacts with the others through a coupling term given by the sine of their phase differences. For high values of the coupling parameter $C$, they shift from independent motion to a synchronised state. A practical measure derived from this model is the Kuramoto Order Paramerter (KOP):

$$R(t) = \frac{1}{r} \left| \sum_{k=1}^{r} e^{i\phi_k(t)} \right|. \tag{2}$$

It measures the synchronisation state at time point $t$. This parameter lies between $0$ and $1$, where $1$ indicates a perfect synchronisation. This would mean, that all oscillators have identical phase values. The Kuramoto model as well as the KOP have been adopted for clustering (Böhm et al., 2010). The data points themselves are viewed as the interacting units - the oscillators - and the synchronisation of every coordinate of each data points is changed due to an iteration rule derived from the Kuramoto model. The KOP is used as a control measure for the state of synchronisation at each iteration. As soon as it is close enough to $1$ the data points are viewed as synchronised and the algorithm stops. We took this perspective as an inspiration for our synchronisation-based cluster loss. However, instead of simulating the synchronisation behaviour of the data points according to the Kuramoto model and observing the state of synchronisation with the KOP, we developed a new cluster loss function $\mathcal{L}_{clu}$ based on the KOP, such that the AE aims at synchronisation. In Appendix A.10 we give more details about the Kuramoto model, the KOP and how they were previously adapted for clustering.

Our intuition is that objects should synchronise stronger, i.e., align their positions, if they are close to each other, since this indicates strong similarity. Their attraction should, furthermore, be amplified if they are assigned to the same cluster. In contrast, points with different labels should not align their positions. The following synchronisation loss $\mathcal{L}_{syn}$ incorporates this intuition for a mini-batch $\mathcal{B}$:

$$\mathcal{L}_{syn}(\mathcal{B}) = 1 - \frac{1}{|\mathcal{B}|^2} \sum_{x \in \mathcal{B}} \sum_{y \in \mathcal{B}} e^{-||z_y - z_x||_2^2 \cdot w_x(y)} \tag{3}$$

The exponential function $f(dist) = e^{-dist}$ monotonically increases with decreasing increment $dist$, with a maximum value of $1$ (considering only non-negative increments). Hence, as embedded points $z_x$ and $z_y$ move closer to each other, the normalised sum of exponential terms in Equation (3) converges to $1$, causing $\mathcal{L}_{syn}$ to approach its minimum of $0$. However, we do not want all points to be drawn to each other to the same extent. Hence, we include the weight term $w_x(y) \in [0, 1]$ to ensure that points with the same label synchronise strongly, points with different labels do not synchronise, and unlabelled points interact with all other points in the batch depending on their distance:

$$w_x(y) = \begin{cases} 1, & \exists i : x, y \in C_i \\ 0, & \exists i, j : i \neq j, x \in C_i, y \in C_j \ , \\ \alpha_x(y), & \nexists i : x \in C_i \end{cases} \quad \text{with} \quad \alpha_x(y) = e^{-\left(\frac{||z_y - z_x||_2^2}{\max_{u \in \mathcal{B}} ||z_x - z_u||_2^2}\right)^2} .$$

The $\alpha \in (0, 1)$ term ensures that nearby points $y$ receive a high weight close to $1$, while distant neighbours receive weights near $0$, with a gradual decay based on distance. The loss $\mathcal{L}_{syn}$ aims to compress the data. Hence, for training the network we need to include a regularisation term, that prevents the embedded space from collapsing by mapping all points to a single point. We use the reconstruction loss $\mathcal{L}_{rec}$ defined in the beginning of Section 3 for that, such that our final loss is

$$\mathcal{L}(\mathcal{B}) = \mathcal{L}_{rec}(\mathcal{B}) + \mathcal{L}_{syn}(\mathcal{B}). \tag{4}$$

Training terminates automatically once all data points have been assigned high-confidence labels (cf. Section 3.3) or when the number of labelled points remains constant over $T_{stop}$ successive epochs.

### 3.3 Gradual Assignment Strategy

After pre-training, the core points are assigned to a cluster. Their label does not change any more during training. We refer to their labels as *high-confidence* labels. During training, the core regions get more compressed and unlabelled points on the border are drawn closer to the high-confidence label region. After each epoch, the whole data set is embedded, and we consider each unlabelled point's $k_{neigh}$ nearest neighbours, including points with label $-1$ ('unlabelled'). A majority vote among the labels of these neighbours then decides on the label of the respective unlabelled point, as shown in Figure 3. This defined label is of low confidence, which means it can change again in the following epoch. Once a label stays the same for $T_{conf}$ epochs, its status changes to high-confidence. We then consider this point a core point. Importantly, once assigned to a cluster, a point can only switch to another cluster, but not be unlabelled any more.

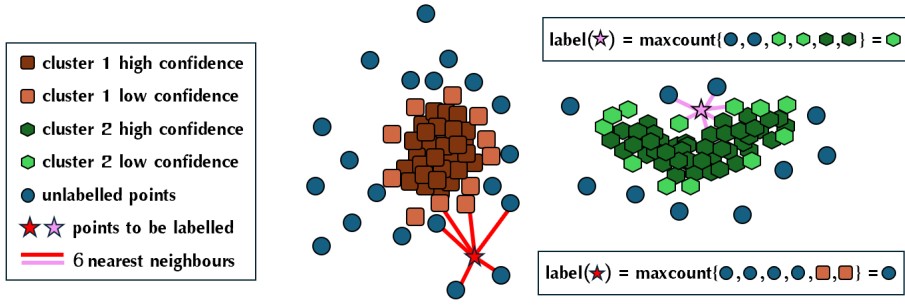

Figure 3: Visualisation of our gradual labelling strategy. After every epoch, unlabelled points as well as points with low-confidence labels are assigned based on their nearest neighbours.

## 4 EXPERIMENTS

We evaluate our approach beginning with quantitative experiments in Section 4.1. We compare the clustering performance of DeepSynC on a series of benchmark data sets to relevant baselines from interaction-based clustering (Mean Shift, Affinity Propagation and SynC) and Deep Clustering (DEC, IDEC, DCN, ACe/DeC, DipDECK), which are described in Section 2. We further analyse the number of clusters detected by DeepSynC, conduct ablation studies on algorithm-specific parameters, and demonstrate the impact of our gradual assignment strategy and the resulting automatic stopping criterion. In Section 4.2, we qualitatively assess the number of clusters found as well as the unassigned points.

**Experimental Setting.** We fix the AE dimensions to $d$-265-128-64-$m$-64-128-256-$d$. The minimum $m = \min\{d, 10\}$ for the embedded space dimension is only relevant for the low-dimensional synthetic data sets used for visualisations. We use the ADAM optimiser and a constant learning rate of 0.001 for the pre-training as well as 0.0001 for the clustering process and a batch size of 256. With this setting, we pre-train 5 AEs for each data set for 100 epochs, which we then use for all further evaluations. The number of epochs for the clustering procedure is set to 150 for our competitors. For all deep clustering competitors, the implementations from the Python package ClustPy (Leiber et al., 2023) were used with the algorithm-specific default settings which are recommended in the respective papers. An exception is DipDECK, which needs an overestimation of the ground truth number of clusters $k$. We use $2 \cdot k$, as the default of 35 does not fulfil this for some data sets.

For DeepSynC, we set $T\%$, which determines the reference area to define the core points, to 0.1, $k_{neigh}$ to 25, and both $T_{conf}$ and $T_{stop}$ to 3. The clustering epochs are automatically determined. The implementation is based on PyTorch (https://pytorch.org/) and is available at: https://anonymous.4open.science/r/DeepSynC-6C21/README.md.

For $k$-Means, MeanShift, and Affinity Propagation, the *scikit-learn* Python implementations were used with the default settings. For both SynC and SHiP we used the Python implementation provided by the authors with the same (default) settings for all data sets. All methods that require the number of clusters as an input parameter are given the ground truth number of classes. The experiments were conducted on a server with 2×Intel 6226R CPUs, each with 16 cores running at 2.9GHz, with 512GB of RAM.

**Evaluation Metric.** We evaluate the quality with the adjusted rand index (ARI) (Hubert & Arabie, 1985), where higher is better. In Table 2, the ARI values are multiplied by 100 for clarity. DeepSynC leaves data points unassigned (labelled with $-1$), if they do not synchronise with the core regions. While this is desirable for noisy real-world data, it is not straightforward how to compare with methods that assign all points. For a fair comparison, we report the ARI results for DeepSynC based on the labelled points together with the percentage of these points. We additionally report the ARI when each unlabelled point is considered as a singleton cluster as DeepSynC+.

**Data Sets.** We conduct experiments on various kinds of benchmark data sets, displaying different properties. We include the image data sets USPS (Hull, 1994), MNIST (LeCun et al., 1998),

Table 1: Core point clustering (core) vs. clustering all embedded points (all) after pre-training. Some results marked with [1] were obtained on $10\%$ of the data due to full-run failures for our competitor.

| Method | | MNIST | FMNIST | USPS | Optdigits | HAR | HTRU | MICE | Pendigits |
|---|---|---|---|---|---|---|---|---|---|
| SHiP | core | $0.89 \pm 0.05$ | $0.54 \pm 0.03$ | $0.93 \pm 0.03$ | $0.92 \pm 0.03$ | $0.61 \pm 0.06$ | $0.58 \pm 0.10$ | $0.41 \pm 0.05$ | $0.93 \pm 0.01$ |
| | all | $0.64 \pm 0.04$ | $0.32 \pm 0.02$ | $0.33 \pm 0.07$ | $0.73 \pm 0.04$ | $0.47 \pm 0.03$ | $0.00 \pm 0.00$ | $0.14 \pm 0.04$ | $0.70 \pm 0.02$ |
| SynC | core | $0.85 \pm 0.06^1$ | $0.42 \pm 0.09^1$ | $0.79 \pm 0.04$ | $0.85 \pm 0.05$ | $0.59 \pm 0.04$ | $0.42 \pm 0.20$ | $0.45 \pm 0.05$ | $0.81 \pm 0.03$ |
| | all | $0.44 \pm 0.07^1$ | $0.25 \pm 0.04^1$ | $0.22 \pm 0.11$ | $0.68 \pm 0.02$ | $0.38 \pm 0.06$ | $0.50 \pm 0.06$ | $0.13 \pm 0.02$ | $0.38 \pm 0.06$ |
| $k$-Means | core | $0.75 \pm 0.02$ | $0.53 \pm 0.02$ | $0.80 \pm 0.03$ | $0.87 \pm 0.01$ | $0.68 \pm 0.05$ | $0.41 \pm 0.33$ | $0.40 \pm 0.06$ | $0.88 \pm 0.02$ |
| | all | $0.65 \pm 0.02$ | $0.40 \pm 0.02$ | $0.56 \pm 0.01$ | $0.68 \pm 0.02$ | $0.57 \pm 0.03$ | $0.57 \pm 0.02$ | $0.20 \pm 0.02$ | $0.57 \pm 0.00$ |

Table 2: Average ARI ($\pm$ std. deviation) and percentage of labelled points (LP) on the 5 pretrained models. DS+= DeepSynC+, S=SynC, MS=MeanShift, AP=Affinity Propagation, DipD=DipDECK. [1] SynC was performed on $10\%$ of the data due to full-run failures.

| Data Set | LP | Ours | DS+ | AE+S | AE+MS | AE+AP | DEC | IDEC | DCN | DipD | ACe/DeC | DKM |
|---|---|---|---|---|---|---|---|---|---|---|---|---|
| **MNIST** | 99.04 | **75±02** | 74±02 | 44±07¹ | 00±00 | 02±00 | 63±08 | 68±03 | 59±06 | 35±06 | 61±02 | 55±05 |
| **FMNIST** | 97.40 | 40±02 | 39±02 | 25±04¹ | 00±00 | 02±00 | 45±03 | **46±03** | 41±01 | 43±02 | 41±01 | 42±03 |
| **USPS** | 95.64 | **75±05** | 72±04 | 22±11 | 00±00 | 10±00 | 70±04 | 72±03 | 61±05 | 72±06 | 54±02 | 67±04 |
| **Optdigits** | 97.86 | **88±02** | 86±02 | 68±02 | 00±00 | 12±01 | 74±05 | 71±06 | 66±06 | 65±18 | 62±05 | 61±03 |
| **COIL20** | 59.33 | **71±06** | 49±03 | 40±05 | 00±00 | 41±05 | 60±02 | 60±02 | 59±02 | 21±18 | 04±07 | 57±03 |
| **COIL100** | 53.68 | **79±06** | 53±02 | 02±00 | 00±00 | 48±02 | 60±01 | 58±01 | 55±02 | 00±00 | 00±00 | 54±01 |
| **Pendigits** | 84.73 | **84±02** | 72±01 | 38±06 | 00±00 | 11±00 | 66±05 | 61±01 | 61±04 | 73±01 | 61±04 | 58±07 |
| **HAR** | 99.55 | 46±05 | 46±05 | 38±06 | 33±00 | 05±00 | 62±02 | **65±07** | 57±08 | 51±00 | 47±04 | 51±00 |
| **HTRU** | 90.40 | 44±13 | 59±02 | 50±06 | 43±02 | 00±00 | 03±00 | 03±00 | 47±01 | 11±21 | 60±06 | **62±04** |
| **Letter** | 85.26 | 22±03 | 18±02 | 01±01 | 00±00 | 08±00 | 21±01 | 19±01 | 18±02 | 01±01 | 23±02 | 06±01 |
| **MICE** | 95.04 | **23±03** | 22±02 | 13±02 | 00±00 | 13±01 | 19±05 | 20±05 | 19±04 | 14±03 | 14±04 | 20±01 |
| **Weizm.** | 81.71 | 34±02 | 30±02 | 05±02 | 01±01 | 30±01 | **37±01** | 35±01 | 35±01 | 01±00 | 07±09 | 34±01 |

Fashion-MNIST (FMNIST) (Xiao et al., 2017), and Optdigits (Markelle Kelly, 2023). Further, we include the tabular data sets Pendigits, Letterrecognition (Letter), HAR, and MICE from the UCI repository (Markelle Kelly, 2023). Finally, we also include the image data sets COIL20 (Nene et al., 1996b) and COIL100 (Nene et al., 1996a) and the video data set Weizmann (Blank et al., 2005). Especially the latter data sets are very challenging as they do not follow a Gaussian distribution cluster assumption. We give further details on the data sets and their preprocessing in Section A.2.

## 4.1 QUANTITATIVE EXPERIMENTS

**Gradual Cluster Assignment.** DeepSynC first only clusters core points instead of assigning all embedded data after pre-training. As shown in Table 1, this improves clustering performance: ARI scores are consistently higher when clustering core points compared to the full dataset. We show this with $k$-Means, SynC, and SHiP on eight datasets (remaining data sets in Section A.3 Table 5), observing the same pattern throughout—except for HTRU, where SynC and $k$-Means perform better on the full set. Since SynC does not necessarily label all points, for fairness, ARI values are computed on the labelled points only. However, for all used data sets SynC did label at least $98.99\%$ of the data.

**Cluster Performance and Percentage of Assigned Points.** Table 2 shows the average ARI results for DeepSynC and our competitors. On most data sets DeepSynC is best or second best (9 out of 12 cases). Even for the unfavourable evaluation method (DeepSynC+), the method is second best to DeepSynC 4 times. For most data sets, we assign a large percentage of points before the algorithm automatically stops (column LP), with consistently low standard deviations below $0.05$ for all data sets. While desirable here, in other data it may be important not to assign ambiguous points.

**Estimated Number of Clusters.** DeepSynC does not require users to pre-define the number of clusters. This is also the case for DipDECK, Affinity Propagation, MeanShift, and SynC. Cluster number estimation is not a primary goal in this paper. Nevertheless, DeepSynC estimates the number of clusters closest or second closest in 8 of 12 cases, as shown in Table 6 in Section A.4.

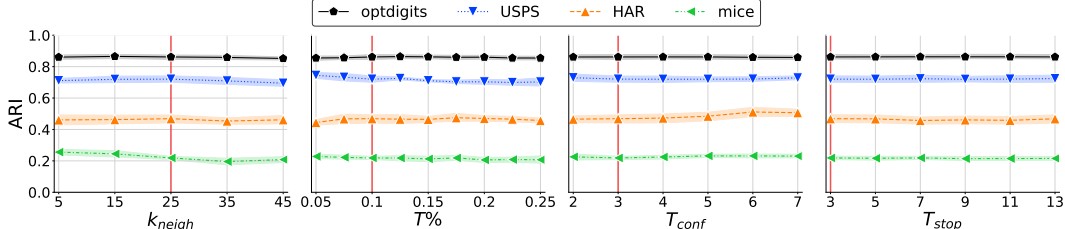

Figure 4: Ablation study. ARI results for four data sets, when varying one parameter $k_{neigh}$, $T\%$, $T_{conf}$, $T_{stop}$ and keeping the others at the default value (vertical red lines). The markers show the mean across all 5 models, the coloured band is the $75\%$ confidence interval.

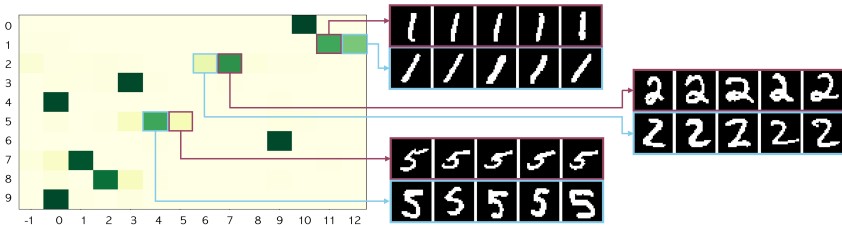

Figure 5: MNIST substructures. In the confusion matrix, the $x$-axis corresponds to our predicted labels, the $y$-axis to ground truth labels. The three additional clusters found by DeepSynC on MNIST show two clusters of '1' (straight vs skewed), '2' (with or without loop), and '5' (skewed vs straight).

**Ablation Studies.** Training of DeepSynC automatically stops when one of two stopping criteria is reached: 1) each point is assigned to the same cluster for $T_{conf}$ epochs, or 2) no new points are assigned for $T_{stop}$ consecutive epochs. We investigate their influence on the clustering performance by varying $T_{conf}$ within $\{2, 3, 4, 5, 6, 7\}$ and $T_{stop}$ within $\{3, 5, 7, 9, 11, 13\}$. Further, we use the $k_{neigh}$ parameter to determine the core points as well as for the label assignment. We investigate the effect of varying $k_{neigh}$, by applying DeepSynC with $k_{neigh} \in \{5, 15, 25, 35, 45\}$. The only remaining parameter is $T\%$, which determines the reference area for a point to decide about its status (core vs. non-core). We investigate $T\%$ between $0.05$ and $0.25$ in $0.025$ steps. Whenever varying one parameter, we keep the other parameters at the default setting $[k_{neigh}, T\%, T_{conf}, T_{stop}] = [25, 0.1, 3, 3]$. The results of these ablation studies are shown in Figure 4.

### 4.2 QUALITATIVE EXPERIMENTS

**Predicted Clusters.** Our $k$-estimation is competitive across all data sets. For the image data sets MNIST, FMNIST, and Optdigits, DeepSynC tends to slightly overestimate the number of clusters compared to the ground truth. This can, however, reveal interesting substructures, e.g., different writing styles of the digits in MNIST (cf. Figure 5). Similar results can be found for Optdigits in Section A.6, as well as different styles of the same category of fashion items in FMNIST.

**Inspection of Unassigned Points.** DeepSynC assigns points to clusters to a large extent, but leaves ambiguous points unassigned. Figure 6a shows randomly selected Optdigits data points in this category. From a visual inspection, this is intuitive, as they are hard to assign to any cluster, and the ground truth cluster often does not correspond well to human perception.

The class 'bags' in FMNIST shows a large variety in styles, as the images in Figure 6b all belong to this class. DeepSynC seems to detect a certain substyle of squared bags without a handle (left). The large variation in styles within this class leads to leaving the images on the right unassigned.

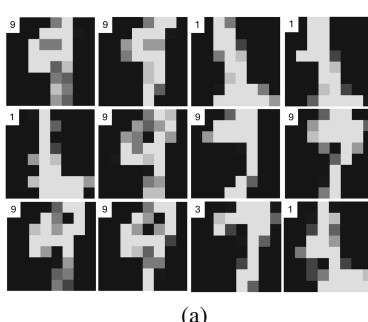 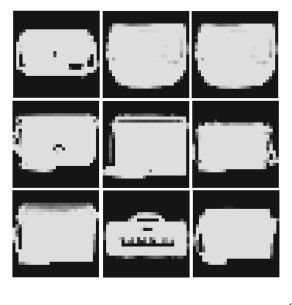 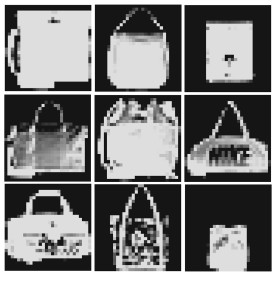

(a)                              (b)

Figure 6: (a) Randomly selected Optdigits data points not assigned by DeepSynC. Numbers in the top-left corner indicate ground truth labels. (b, left) Randomly selected images with the same predicted label by DeepSynC, and ground truth 'bag'. (b, right) Randomly selected images with ground truth label 'bag' that were not assigned by DeepSynC.

### 4.3 DISCUSSION

**Core Points.** Our local core point definition has several desired properties. First, the clusters are well represented by the core points. The number of unique ground truth labels present in the core points is equal to the ground truth number of clusters in all but three data sets: COIL 20, COIL100, and Weizmann, where on average the core points covered $17.2$ (of $20$), $71.4$ (of $100$), and $76.4$ (of $90$) clusters, respectively. This is not surprising, as these data sets display chain-like cluster structures, which is not perfectly aligned with our local density core point definition. Further, clustering the core points achieves better performance compared to clustering all embedded points at once (cf. Table 1). One exception is the $k$-Means and SynC results on the data set HTRU.This is likely due to the significant imbalance between the two clusters ($16,259$ vs. $1,639$). This extreme imbalance is reflected as well in the presence of core points in each cluster (on average $4,130.4$ vs. $87.6$ across the 5 models). In such cases, it would be best to lower the value for $T\%$.

**Automatic Stopping.** The gradual cluster assignment strategy enables the algorithm to decide when to stop training automatically. On average ($113.23 \pm 26.45$), we need less epochs than the standard setting for the competitors ($150$). For Optdigits, for instance, we achieve the best ARI result with only $37.2 \pm 8.9$ epochs. In Section A.5, we report for each data set, the number of epochs for DeepSynC (Table 7). We further analyse the importance of this hyperparameter with additional experiments. First, we vary the number of epochs to see the effect on the performance of our competitors (Figure 9). We then also compare the results of all deep clustering competitors, when for each data set we use the exact same number of epochs as DeepSynC used (Table 8).

**Cluster Performance.** The results in Table 2 show that DeepSynC is competitive across several categories of deep clustering benchmark data sets: images, tabular, and video data. In many cases, we outperform our competitors despite being at a disadvantage of not knowing the ground-truth number of clusters. As expected, our performance drops once we assign each identified outlier as a single cluster. This emphasises that the identified points are indeed the ones that are harder to cluster. Note, however, that it is not our goal to cluster all points, and the results for DeepSynC+ are a necessary makeshift to address the fact that other methods label all points.

**Parameter Robustness.** DeepSynC reduces the number of hyperparameters by eliminating the need for a predefined cluster count and employing an automatic stopping rule. Nevertheless, it introduces new parameters: $k_{neigh}$ and $T\%$ for core point detection, and $T_{conf}$ and $T_{stop}$ as stopping criteria. In extensive ablation studies, we investigate the robustness of our parameters by varying each of them in a certain meaningful range while keeping the others at the default setting. Figure 4 shows that the ARI results remain stable across data sets and parameter variations. Note, that this statement holds for a variety of data sets with different features and different overall performances of DeepSynC: images (Optdigits, USPS) and tabular data (MICE, HAR); data, where we outperform competitors with strong (Optdigits, USPS) and weaker (MICE) cluster performance; data where we are competitive, but rank in the back (HAR); and data with very accurate (Optdigits, USPS, MICE)

and slightly worse (HAR) cluster number estimation. This evaluation verifies the robustness of our parameters.

**Predicted Clusters and Unassigned Points.** DeepSynC is not forced to assign all data points to clusters or fit the data into a certain number of clusters. While this flexibility might lead to unexpected results, it can help interpret the algorithm's behaviour. In general, DeepSynC provides a cluster number estimation that is competitive with the state-of-the-art in deep clustering (Leiber et al., 2021). In some cases, a larger number of clusters reveal interesting substructures, such as handwriting style differences (e.g., MNIST cf. Figure 5 and Optdigits cf. Figure 10). Similarly, variations in boot styles also result in subclusters (cf. Figure 10).

Analysing the unassigned points can be insightful as well. Some of the unassigned points have straightforward justification because they are difficult to recognise even by humans (cf. Figure 6a). However, in other cases, they might look easy to identify, such as the unassigned bags in FMNIST (cf. Figure 6b (right)). Not only are they of largely varying styles – they are mainly different from what the algorithm defines as a 'standard bag' where width is larger than height and most bag pixels are of high intensity (cf. Figure 6b (left)). Moreover, the used network architecture is not advantageous for learning shapes, as we do not use convolutional layers, for instance. This makes it harder for the model to generalise the concept of a bag in the illustrated example. Prior knowledge may simplify the clustering task; however, leaving it unconstrained allows the algorithm to discover what it considers a sufficient structure without compelling unjustified assignments.

For COIL20 and COIL100, the number of unassigned points at the end of the training is low. They consist of images of objects from different angles and therefore display sparse chains of data points in the embedding. This, along with the low number of data points per cluster, inhibits the synchronisation process and consequently the gradual label spreading. A possible solution would be to use a much larger batch size to increase the number of representatives per cluster in each training batch.

## 5 CONCLUSION

In this paper, we introduce the first deep synchronisation-based clustering method, DeepSynC. We differentiate between points that are unambiguously belonging to a cluster, so-called core points in the dense area of a cluster, and points that are more difficult to assign. This allows the generation of an embedding that is very suitable for clustering, as the autoencoder is not misled by initial fuzzy borders between clusters. Employing our novel synchronisation loss allows us to find clusters of non-convex shapes with unclear boundaries in very high-dimensional data, such as image data. The concept of core points opens pathways for future work, e.g., using it for outlier detection.

## REPRODUCIBILITY STATEMENT

This work includes an anonymous git repository (`https://anonymous.4open.science/r/DeepSynC-6C21/README.md`) including the code to run and reproduce the experiments. Hyperparameters for competitors, as well as the data, including data processing steps, are either included in the code or referenced.

## ETHICS STATEMENT

This work aims to uphold the ethical conduct of research in accordance with established computing ethics guidelines. We focus on foundational data mining research, striving for scientific output that contributes positively to new research discoveries in our field, but also in application domains. To facilitate reproducibility and maximise transparency, we provide our code and document all necessary parameters. All used data sets are open-source and cited accordingly.

The method presented in this paper is a generally applicable clustering algorithm, and while no sensitive data, such as medical records or unanonymised data, were involved in the development or evaluation of our method, there is potential for misuse in sensitive applications. For example, our method might be used to cluster medical data and identify patients displaying a different pattern of features ('unassigned points'). The utmost caution must be exercised when drawing conclusion

about human health conditions, as we cannot exclude the possibility of incorrect information due to inappropriate usage or algorithmic errors. Therefore, we encourage users to consider the ethical implications of their specific applications.

## LLM USAGE STATEMENT

In some paragraphs, we used LLMs as a post-processing step to improve wording and grammar. While we did not copy anything above sentence level, we drew inspiration for shortening or phrasing more elegantly. Text, figures, and content of the paper are our own work and have **not** been generated, updated, or processed with LLM usage.

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

# A APPENDIX

## A.1 NOTATION

Table 3 gives an overview of the used notation in the mathematical formulas.

Table 3: Explanation of used notations. (dim. = dimensionality; # = number of)

| Notation | Meaning | Notation | Meaning |
|---|---|---|---|
| $d \in \mathbb{N}$ | dim. original space | $\mathcal{C}$ | set of cluster labels |
| $m \in \mathbb{N}$ | dim. embedded space | $C_i \subseteq \mathcal{D}, i \in \mathcal{C}$ | objects assigned to cluster $i$ |
| $k \in \mathbb{N}$ | #clusters | $\mathcal{B} \subseteq \mathcal{D}$ | a mini-batch |
| $n \in \mathbb{N}$ | #points in data set | $x \in \mathcal{D}$ | a single object of the data set |
| $b \in \mathbb{N}$ | #points in mini-batch | $\hat{x} \in X$ | reconstruction of $x$ ($\hat{x} = dec(enc(x))$) |
| $\mathcal{D} \subseteq \mathbb{R}^d$ | set of all objects | $z_x \in enc(\mathcal{D})$ | object in embedded space ($z_x = enc(x)$) |

## A.2 DATA SET DETAILS

Table 4 gives an overview of the data sets we used, including their characteristics and sources. We used famous real-world benchmark data sets for tabular and image classification. All image and video data sets are reshaped into a one-dimensional vector and preprocessed by a channel-wise z-transformation. The tabular data sets are preprocessed using a feature-wise z-transformation.

Table 4: Data set properties. Number of samples ($n$), dimensions ($d$), number of ground truth clusters ($k$), and the source.

| Data Set | $n$ | $d$ | $k$ | Source |
|---|---|---|---|---|
| HTRU | 17,898 | 8 | 2 | Markelle Kelly (2023) |
| Pendigits | 10,992 | 16 | 10 | Markelle Kelly (2023) |
| MICE | 1,077 | 68 | 8 | Markelle Kelly (2023) |
| Letterrec | 20,000 | 16 | 26 | Markelle Kelly (2023) |
| HAR | 10,299 | 561 | 6 | Markelle Kelly (2023) |
| Optdigits | 5,620 | 64 | 10 | Markelle Kelly (2023) |
| USPS | 9,298 | 256 | 10 | Hull (1994) |
| MNIST | 70,000 | 784 | 10 | LeCun et al. (1998) |
| FMNIST | 70,000 | 784 | 10 | Xiao et al. (2017) |
| COIL20 | 1,440 | 16,384 | 20 | Nene et al. (1996b) |
| COIL100 | 7,200 | 49,152 | 100 | Nene et al. (1996a) |
| Weizmann | 5,701 | 77,760 | 90 | Blank et al. (2005) |

### A.3 LOCAL VS. GLOBAL CORE POINTS

**Motivation for T%.** In this section, we illustrate the motivation for introducing the parameter $T\%$ to calculate local instead of global, i.e., $T\% = 1.00$, core points. Figure 7 visualises the effect of varying $T\%$ and $k_{neigh}$ on a toy data set. The impact of $T\%$ is minor for quite a range $[0.05, 0.1, 0.25]$. However, when considering all data points as a reference, i.e. $T\% = 1.00$, the clusters are not represented well anymore in the now 'global' core points irrespective of $k_{neigh}$. This is most prominent in the big, yellow cluster. This motivates us to consider local core points.

**Local Core Points vs. DBSCAN core Points.** DBSCAN does also have a core points notion. The goal of both core-type notions, the one of DeepSynC as well as the one of DBSCAN is to identify points that are central points of a cluster. While both approaches rely on a notion of local density, i.e., core points are expected to have higher local density than other points, there are important conceptual and practical differences. DBSCAN depends on a global density threshold, which makes its core-point definition highly sensitive to parameter choices. In particular, DBSCAN may fail to identify any core points in the dataset when minPts is set too high or when $\varepsilon$ is chosen too small. Choosing $\varepsilon$ too large can cause that all points including clear outliers or noise are labeled as core points. Moreover, DBSCAN's fixed global threshold is inherently unsuitable for datasets containing clusters with different densities, a scenario that is common in real-world applications. In Figure 8, we illustrate the effects of the two DBSCAN parameters $\varepsilon$ and $min_p ts$ on the same toy data set that we used in Figure 7 to investigate our parameters. The above mentioned issues can clearly be seen, even for a quite simple synthetic data set. In contrast, our definition adapts naturally to both the dataset and the individual clusters. Because DeepSynC relies on the k-neighborhood rather than a hard-to-tune $\varepsilon$ value, it always yields core points. By comparing relative local densities, DeepSynC robustly identifies core points within each cluster across a wide range of values for both $T\%$ and $k_{neigh}$. This leads to more stable and meaningful core-point detection, especially in real world datasets.

**Core Point Clustering vs. Clustering all Embedded Points.** In the main paper in Section 4.1 Table 1, we present the clustering results in the core points compared to the clustering results, when considering all embedded points for eight of the twelve used benchmark data sets. Table 5 contains the results for the remaining four data sets. We can see that the pattern is the same as for other data sets: Clustering core points yields better results than clustering all data points.

### A.4 CLUSTER NUMBER ESTIMATION

Although the specific cluster number estimation task is not a primary goal in this paper, our algorithm does have the benefit that the cluster number does not have to be specified in advance. In

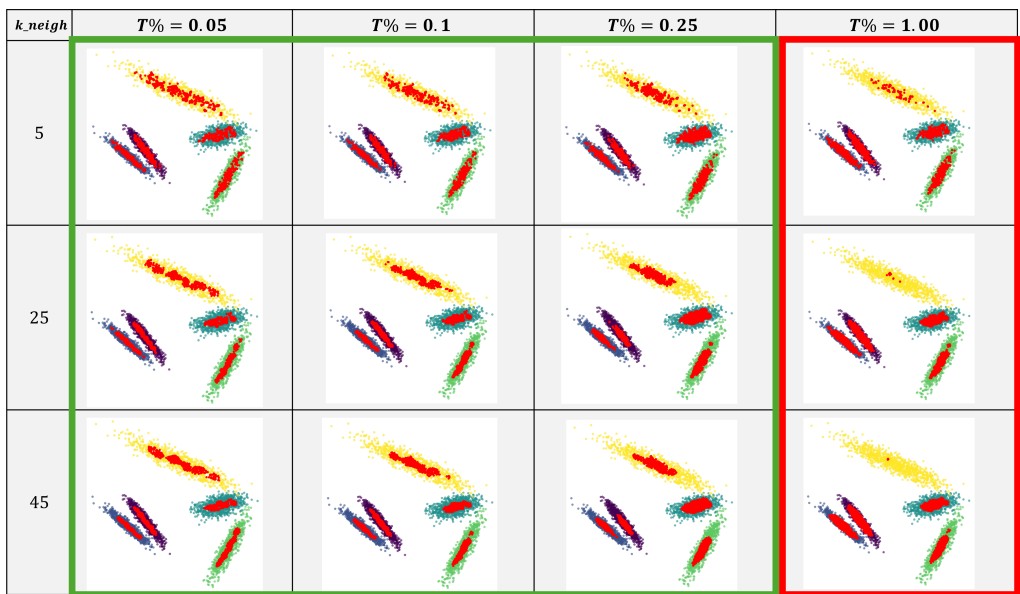

Figure 7: Visualisation of the effect of the parameters $T\%$ and $k_{neigh}$. The red data points are the determined core points for this 2-dimensional data set with 5 ground truth clusters. In each figure, $T\%$ and $k_{neigh}$ are set differently as indicated in the header and the first column respectively.

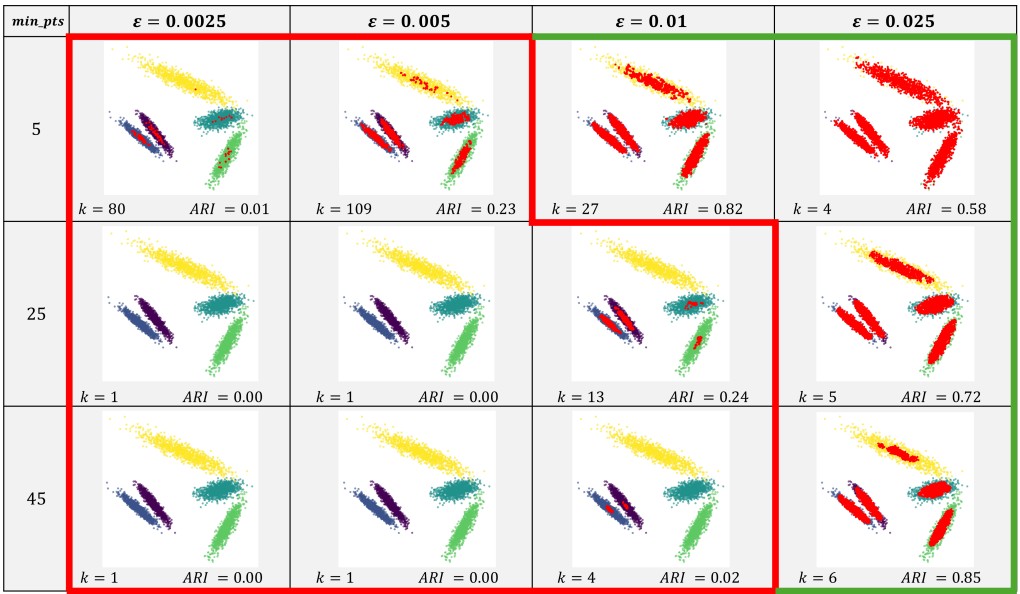

Figure 8: Visualisation of the effect of the parameters $\varepsilon$ and $min_p ts$. The red data points are the points that DBSCan identifies as core points for this 2-dimensional data set with 5 ground truth clus-ters. In each figure, $\varepsilon$ and $min_p ts$ are set differently as indicated in the header and the first column respectively.

Table 5: Core point clustering (core) vs. clustering all embedded points (all) after pre-training for the remaining four data sets. The other 8 dat sets are shown in Table 1 in the main paper.

| Method | | | COIL20 | COIL100 | Letter | Weizmann |
|---|---|---|---|---|---|---|
| SHiP | core | | $0.89 \pm 0.03$ | $0.91 \pm 0.02$ | $0.33 \pm 0.01$ | $0.63 \pm 0.01$ |
| | all | | $0.78 \pm 0.02$ | $0.70 \pm 0.08$ | $0.18 \pm 0.01$ | $0.45 \pm 0.02$ |
| SynC | core | | $0.82 \pm 0.08$ | $0.24 \pm 0.07$ | $0.09 \pm 0.01$ | $0.18 \pm 0.04$ |
| | all | | $0.40 \pm 0.05$ | $0.02 \pm 0.01$ | $0.01 \pm 0.01$ | $0.05 \pm 0.02$ |
| $k$-Means | core | | $0.86 \pm 0.01$ | $0.78 \pm 0.01$ | $0.29 \pm 0.02$ | $0.67 \pm 0.03$ |
| | all | | $0.60 \pm 0.04$ | $0.57 \pm 0.01$ | $0.18 \pm 0.01$ | $0.39 \pm 0.01$ |

Table 6: k-estimation results. [1] Performed on 10% of the data due to full-run failures.

| Data Set (k) | DeepSynC | SynC | DipDECK | MeanSh. | Aff. Prop. |
|---|---|---|---|---|---|
| MNIST (10) | 13.40±1.50 | 6.20±0.40[1] | **11.20±1.47** | 1.00±0.00 | 1173.4±18.1 |
| FMNIST (10) | 14.00±3.52 | 7.80±0.40[1] | **11.00±1.26** | 1.00±0.00 | 823.6±10.7 |
| USPS (10) | 8.20±0.40 | 8.40±0.80 | **10.40±3.01** | 1.00±0.00 | 263.2±4.8 |
| Optdigits (10) | **11.40±1.02** | 8.80±1.94 | 15.60±0.49 | 1.00±0.00 | 161.4±7.3 |
| COIL20 (20) | 14.80±1.17 | 8.60±3.61 | **24.60±1.74** | 1.00±0.00 | 222.6±45.4 |
| COIL100 (100) | **52.60±3.88** | 7.40±1.62 | 21.40±2.58 | 1.00±0.00 | 1045.6±107.8 |
| Pendigits (10) | **14.20±0.75** | 15.00±1.67 | 16.40±0.49 | 1.00±0.00 | 193.4±7.1 |
| Har (6) | 11.40±2.06 | 3.20±0.40 | **6.80±0.75** | 3.00±0.89 | 199.6±7.0 |
| HTRU (2) | 3.20±0.40 | **1.20±0.40** | 5.40±1.36 | 11.20±1.17 | 439.6±8.7 |
| Letter (26) | 82.60±4.36 | 10.40±2.87 | **12.40±2.42** | 1.00±0.00 | 592.0±38.7 |
| MICE (8) | **5.60±0.80** | 2.60±0.80 | 5.40±1.36 | 1.80±0.40 | 47.6±3.3 |
| Weizmann (90) | **43.20±3.25** | 13.00±4.05 | 17.60±1.85 | 1.60±0.49 | 415.2±165.1 |

Table 6, we report the number of clusters detected by DeepSynC as well as by the competitors that do not require $k$ as an input, on all data sets. DeepSynC is highly competitive with the state-of-the-art in deep clustering $k$-estimation DipDECK (Leiber et al., 2021).

## A.5 EFFECTS OF HYPERPARAMETER '$n_{epochs}$'

**Varying the Number of Epochs.** In our first experiment, we executed the deep clustering competitors on all models of all data sets with a varying number of clustering epochs $n_{epochs} \in \{40, 100, 150, 200\}$. For each data set, we additionally include the number of epochs as determined by DeepSynC. The plots in Figure 9 show the results on a selection of data sets to illustrate the effects that translate to other data sets as well. For each data set, some algorithms show stable performance considering mean and standard deviation across the five models (right hand side figures), while others show largely varying performance (left hand side figures). One exception is DipDECK on HTRU (Figure 9e), which shows very stable performance on average across models. However, the performance is consistently poor, and the standard deviation across models is very large. Importantly, we can observe, that for each algorithm, there is at least one data set, where the influence of the number of epochs on the performance is most evident.

**DeepSynC Number of Epochs.** Table 8 shows the performance results of DeepSynC as well as of our competitors, when we fix the number of training epochs to the number of epochs DeepSynC determined. The ranking of the algorithms per data set does only change in a few cases. For instance, for Weizmann previously DKM was the best-performing, now it is surpassed by DEC and IDEC. In total, the performance improves by at least 0.02 in average ARI (green) in 13 cases and drops at least as much in ARI in 18 cases. There is no clear pattern if this connects to a lower or respective higher number of training epochs, a specific algorithm or a certain data set. Only for Weizmann it seems to be the case that more algorithms benefit from a longer training.

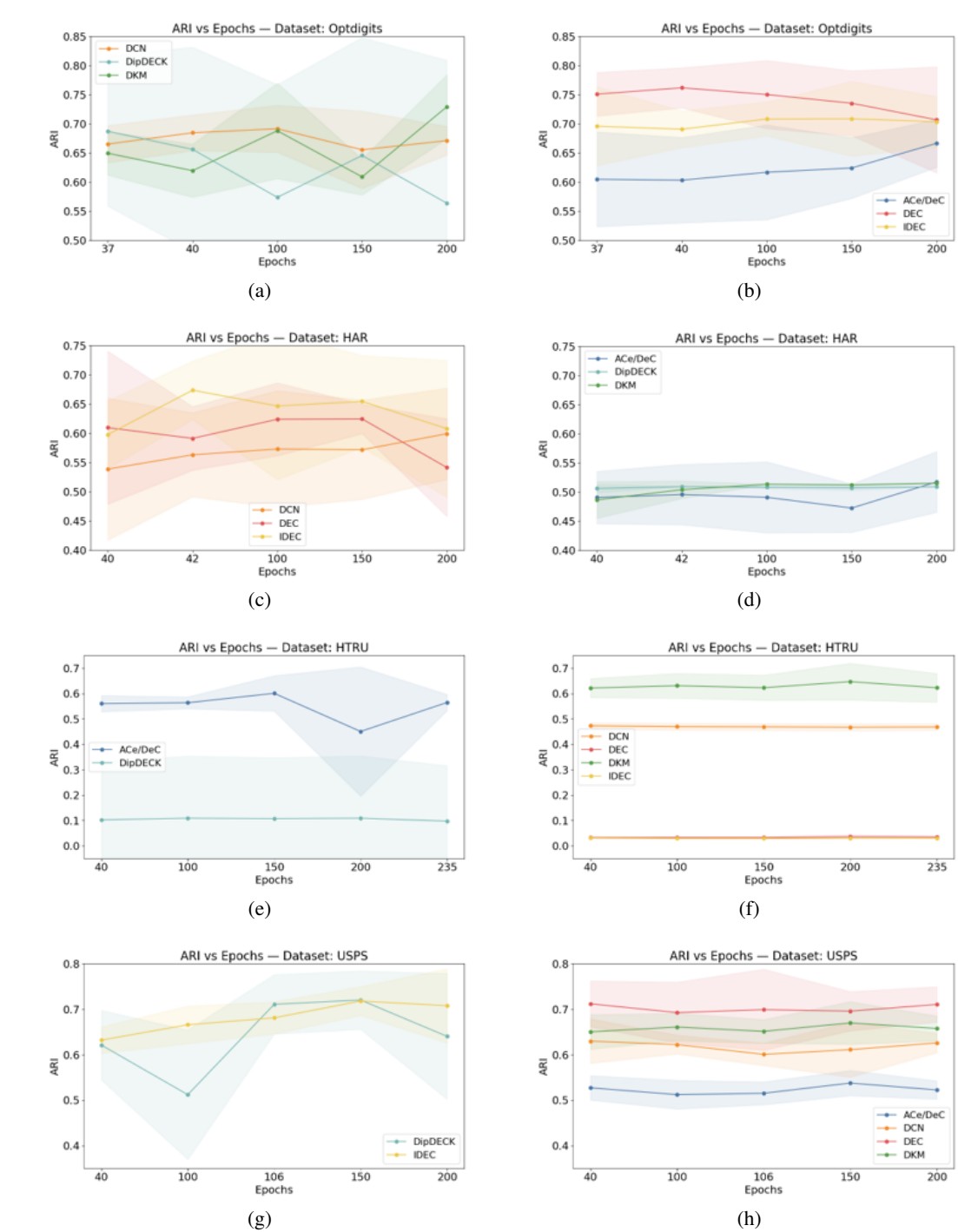

Figure 9: The subfigures on the left (a), (c), (e) and (g) show the performance on the five pre-trained models of a single data set for those deep clustering algorithm that show an unstable performance across varying number of epochs. The subfigures on the right, (b), (d), (f) and (h) show the results of algorithms with stable performance.

These results support the claim that the number of epochs is a crucial hyperparameter and that it is favourable if the algorithm can automatically determine when to stop the training process.

Table 7: This table shows for each of the used data sets, how many epochs DeepSynC was trained before it automatically stopped. Additionally, we report the percentage of labelled points as well as the data set size.

| Data set | MNIST | FMNIST | HTRU | Letter | COIL20 | COIL100 |
|---|---|---|---|---|---|---|
| epochs | $204.60 \pm 54.77$ | $258.60 \pm 65.89$ | $235.00 \pm 42.12$ | $191.80 \pm 79.34$ | $36.20 \pm 6.97$ | $35.40 \pm 5.00$ |
| %labelled pts | $99.04 \pm 0.00$ | $97.40 \pm 0.01$ | $90.40 \pm 0.01$ | $85.26 \pm 0.02$ | $59.33 \pm 0.05$ | $53.68 \pm 0.02$ |
| size n | 70,000 | 70,000 | 17,898 | 20,000 | 1,440 | 7,200 |

| Data set | Weizmann | MICE | Pendigits | HAR | USPS | Optdigits |
|---|---|---|---|---|---|---|
| epochs | $98.20 \pm 18.04$ | $40.20 \pm 8.33$ | $73.60 \pm 26.11$ | $42.00 \pm 7.62$ | $106.00 \pm 22.65$ | $37.20 \pm 8.89$ |
| %labelled pts | $81.71 \pm 0.02$ | $95.04 \pm 0.03$ | $84.73 \pm 0.03$ | $99.55 \pm 0.00$ | $95.64 \pm 0.02$ | $97.86 \pm 0.01$ |
| size n | 5,701 | 1,077 | 10,992 | 10,299 | 9,298 | 5,620 |

Table 8: Average ARI results ($\pm$ std. deviation) on the 5 pretrained models. For each data set, the number of epochs for competitors was set to the average number of epochs determined by DeepSynC for the respective data set. Green/red implies whether the ARI scores increased/dropped compared to fixed default numbers of epochs.

| Data set | # Epochs | DeepSynC | DeepSynC+ | DEC | IDEC | DCN | DipDECK | ACe/DeC | DKM |
|---|---|---|---|---|---|---|---|---|---|
| **MNIST** | 205 | **0.75±0.02** | 0.74±0.02 | 0.68±0.04 | 0.68±0.07 | 0.59±0.03 | 0.33±0.12 | 0.62±0.05 | 0.61±0.08 |
| **FMNIST** | 259 | 0.40±0.02 | 0.39±0.02 | 0.44±0.05 | **0.46±0.05** | 0.42±0.01 | 0.43±0.03 | 0.41±0.02 | 0.43±0.02 |
| **USPS** | 106 | **0.75±0.05** | 0.72±0.04 | 0.70±0.08 | 0.68±0.03 | 0.60±0.02 | 0.71±0.06 | 0.52±0.02 | 0.65±0.02 |
| **Optdigits** | 37 | **0.88±0.02** | 0.86±0.02 | 0.75±0.03 | 0.70±0.06 | 0.67±0.03 | 0.69±0.12 | 0.60±0.07 | 0.65±0.03 |
| **COIL20** | 36 | **0.71±0.06** | 0.49±0.03 | 0.60±0.02 | 0.60±0.03 | 0.57±0.02 | 0.47±0.07 | 0.00±0.00 | 0.57±0.05 |
| **COIL100** | 35 | **0.79±0.06** | 0.53±0.02 | 0.59±0.01 | 0.57±0.01 | 0.56±0.01 | 0.00±0.00 | 0.00±0.00 | 0.53±0.02 |
| **Pendigits** | 74 | **0.84±0.02** | 0.72±0.01 | 0.62±0.06 | 0.63±0.04 | 0.60±0.04 | 0.72±0.02 | 0.58±0.04 | 0.60±0.05 |
| **HAR** | 42 | 0.46±0.05 | 0.46±0.05 | 0.59±0.05 | **0.67±0.04** | 0.56±0.06 | 0.51±0.00 | 0.50±0.05 | 0.50±0.01 |
| **HTRU** | 235 | 0.44±0.13 | 0.59±0.02 | 0.04±0.00 | 0.03±0.00 | 0.47±0.01 | 0.10±0.20 | 0.56±0.03 | **0.62±0.05** |
| **Letter** | 192 | 0.22±0.03 | 0.18±0.02 | 0.20±0.02 | 0.20±0.01 | 0.16±0.01 | 0.02±0.02 | **0.22±0.02** | 0.06±0.03 |
| **MICE** | 40 | **0.23±0.03** | 0.22±0.02 | 0.18±0.03 | 0.18±0.03 | 0.17±0.02 | 0.11±0.07 | 0.12±0.03 | 0.18±0.02 |
| **Weizmann** | 98 | 0.34±0.02 | 0.30±0.02 | **0.40±0.02** | **0.40±0.02** | 0.37±0.02 | 0.02±0.03 | 0.00±0.00 | 0.39±0.01 |

## A.6 ADDITIONAL QUALITATIVE EXPERIMENTS

In this section, we show more qualitative experiment results for the cluster number estimation (cf. Figure 10) and the inspection of unassigned points (cf. Figure 11).

**Overestimation of $k$.** As for MNIST (Section 4.2), the overestimation on FMNIST and Optdigits also reveals interesting substructures. In Figure 10a, we can see that DeepSynC identified two subclasses of 'boot': flat boots (bottom) and boots with heels (top). For Optdigits (Figure 10b DeepSynC identified two subclasses of the digit 1: straight lines (top) and hooked (bottom).

**Inspection of Unassigned Points for MNIST.** Figure 11 shows randomly sampled unassigned points in the MNIST data set. The majority belongs to the classes '2', '4', and '7'. For some of them, it is quite intuitive that they did not get assigned, such as the '2' in the top left image. Others, however, mostly the '7's seem easier to cluster.

## A.7 RUNTIME EXPERIMENTS

Our assignment strategy based on the majority vote of the nearest neighbours introduces a possible computationally intensive overhead compared to our competitors. To evaluate whether this has a significant influence, we calculated the average runtime across the five models for each data set and each of the deep clustering algorithms. The results are shown in Figure 12. For better visualisation, we grouped the data sets in subfigures according to their general computational demand across algorithms. In general, the runtime of DeepSynC is influenced most by three factors: (1) the number of training epochs, which is, in contrast to our competitors, not fixed, but determined automatically and therefore shows varying values across data sets as well as pre-trained models (Table 7); (2) the size of the data set as more data points result in a more time-consuming kNN-based calculation of the core points and majority vote assignment strategy; and (3) the percentage of assigned points at

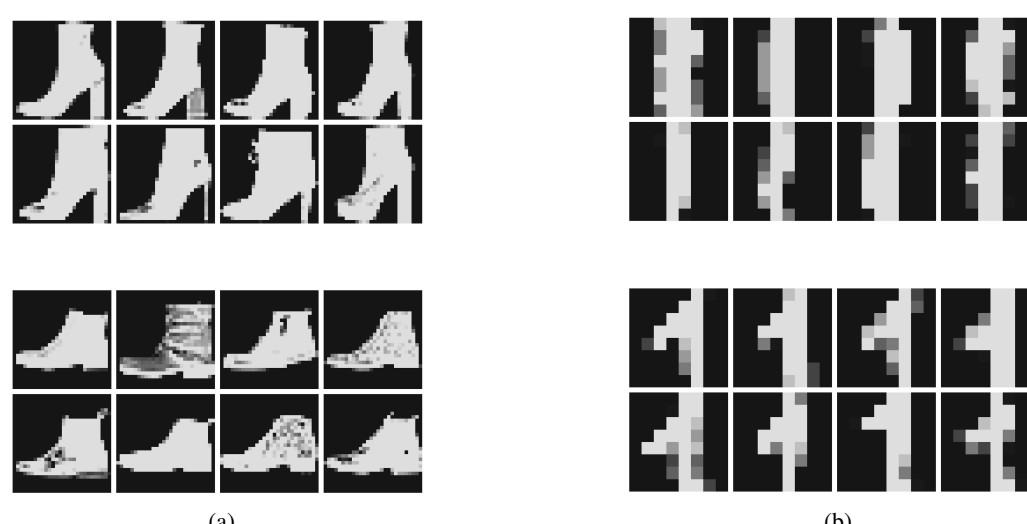

Figure 10: Substructures found in FMNIST (a) and Optdigits (b). DeepSynC identifies two subclasses of 'boot' (heel vs flat) in FMNIST and two subclasses of '1' (straight line vs hook) in Optdigits.

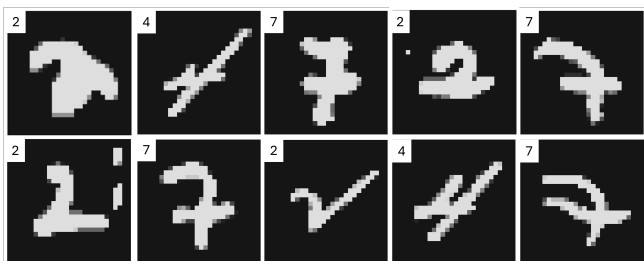

Figure 11: Randomly chosen points that have not been assigned by DeepSynC for MNIST. The numbers in the corner display the respective ground truth labels.

the end of training as a lower number of assigned points indicates a slow label propagation which most likely will result in less training epochs due to our stopping criterion.

Considering these aspects, it is not surprising that DeepSynC has a higher variation in runtime across data sets. For data sets like MNIST, FMNIST, HTRU and Letter DeepSynC is slower than competitors as the number of epochs are way above 150 (204.60±54.77, 258.60±65.89, 235.00±42.12, 191.80±79.34), they are the four largest in size (70,000, 70,000, 17,898, 20,000) and have a quite high percentage of labelled points (99.04±0.00, 97.40±0.01, 90.40±0.01, 85.26±0.02). On others, like COIL20, COIL100, Weizmann and MICE, DeepSynC is among the fastest two as the number of epochs is low (36.20±6.97, 35.40±5.00, 40.20±8.33, 98.20±18.04), MICE and COIL20 are additionally small (1,077, 1,440) and especially COIL20 and COIL100 have a very low percentage of labelled points (59.33±0.05, 53.68±0.02). For the remaining data sets, we are in the midfield as the three factors are not on the extreme end of the spectrum.

We can also observe that DeepSynC has the highest standard deviation across the pre-trained models. This is due to factor (1) as the standard deviation in number of epochs (Table 7) is naturally reflected in a larger standard deviation in runtime.

Judging from the comparisons in Figure 12 and especially Figure 12d, which shows the average runtime across all models and data sets for each method, DeepSynC is slightly slower, but comparable in runtime. The average run for DeepSynC was approximately 12 minutes, while it was about 9 minutes for the second slowest ACe/DeC and 1.5 minutes for the fastest algorithm DEC.

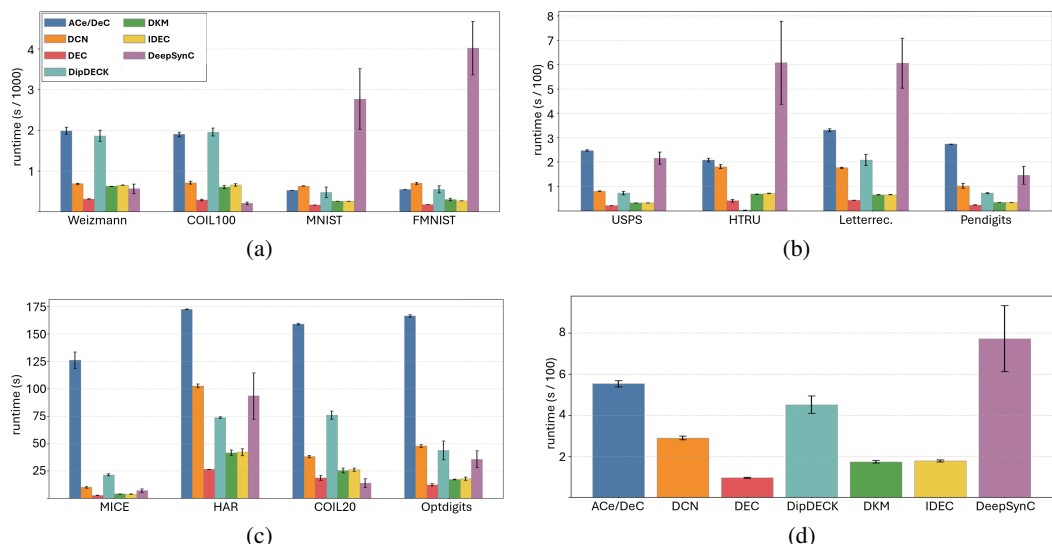

Figure 12: Subfigures (a), (b), and (c) show the average runtime of each deep clustering algorithm across the five pre-trained models for each data set separately. Subfigure (d) shows the average runtime of each algorithm across all models and all 12 data sets.

## A.8 ALGORITHMS

In the following we provide pseudocodes including runtime analysis for all parts of the DeepSynC algorithm. Considering the total time complexity of DeepSynC, selecting the core points calculates the pairwise Euclidean distance, which has the highest runtime complexity of $O(n^2d)$. This runs only once at the beginning of the algorithm. For the sizes of our used datasets, this was faster than using a KD-tree, which would result in a theoretical runtime complexity of $O(nlog(n))$ for this step.

The training loop and gradual assignment have less time complexity; $O(nbd)$ for an epoch and $O(nlog(n))$ for the label assignment strategy, which deals with unlabeled points only. For the gradual assignment strategy KNN is used, which utilises the *kneighbors* method from the scikit-learn implementation. It internally builds a KD-Tree with a time complexity of $O(nlog(n))$. Since this is done after every epoch, the runtime complexity of this part is $O(n_{epochs}nlog(n))$.

Hence, the total complexity is $O(n^2)$. We did not consider approximations, as we did not encounter runtime issues; however, using them could further speed up the algorithm.

---

**Algorithm 1** DeepSynC                                                    Runtime: $\mathcal{O}(n^2 \cdot d)$

---

**Require:** $X, n_{check}, k_p, T\%, n_{epochs}$
1: $model \leftarrow \text{Pretrain}(X)$
2: $Embeddings \leftarrow model.encode(X)$
3: CorePoints $\leftarrow$ Find-Local-CorePoints($Embeddings, k_p, T\%$)                   $\triangleright \mathcal{O}(n^2 \cdot d)$
4: CorePoints-Labels $\leftarrow SHiP(\text{CorePoints})$
5: Initialize the predicted labels ($\hat{y}$) by $-1$ indicating all points are unlabeled.
6: $\hat{y}$ at CorePoints $indices \leftarrow$ CorePoints-Labels
7: **for** $e = 1$ to $n_{epochs}$ **do**                                        $\triangleright \mathcal{O}(n_{epochs} \cdot n \cdot b \cdot d)$
8:    **for** each Batch $B$ in $Batches(X)$ **do**                            $\triangleright \mathcal{O}(n \cdot b \cdot d)$
9:       Compute loss: $\ell \leftarrow \mathcal{L}(B, \hat{y})$
10:       Update model parameters: $\ell.BackPropagate()$
11:    **end for**
12:    $\hat{y} \leftarrow$ Gradual-Label-Assignment($Embeddings, \hat{y}, n_{check}, k_p$)        $\triangleright \mathcal{O}(n \cdot log(n))$
13:    **if** Early-Stopping($\hat{y}$) **then**
14:       Break For Loop
15:    **end if**
16: **end for**
17: **return** $\hat{y}$

---

**Algorithm 2** Find-Local-CorePoints                                       Runtime: $\mathcal{O}(n^2 \cdot d)$

---

**Require:** $Embeddings, k_p, T\%$
1: $n \leftarrow size(X)$
2: $subset \leftarrow floor(n \times T\%)$
3: Pairwise-ED $\leftarrow$ Euclidean-Distance($Embeddings$)                     $\triangleright \mathcal{O}(n^2 \cdot d)$
4: CorePoints-Distances $\leftarrow partition(\text{Pairwise-ED}, k_p, axis = 0)$        $\triangleright \mathcal{O}(n)$
5: NearestNeighbors $\leftarrow argpartition(\text{Pairwise-ED}, subset)$              $\triangleright \mathcal{O}(n)$
6: Median-Thresholds $\leftarrow median(\text{CorePoints-Distances})$                 $\triangleright \mathcal{O}(n)$
7: CorePoints-Mask $\leftarrow$ CorePoints-Distances $<$ Median-Thresholds
8: CorePoints $\leftarrow \hat{y}[\text{CorePoints-Mask}]$
9: **return** CorePoints

---

**Algorithm 3** Gradual-Label-Assignment                                    Runtime: $\mathcal{O}(n \cdot log(n))$

---

**Require:** $Embeddings, \hat{y}, n_{check}, k_p$
1: HighConfidence-$y \leftarrow$ Get-HighConfidence-Labels($\hat{y}, n_{check}$)           $\triangleright \mathcal{O}(n_{check})$
2: $Updated\text{-}\hat{y} \leftarrow \hat{y}$
3: $Updated\text{-}\hat{y}[\sim \text{HighConfidence-}y] \leftarrow -1$
4: Unlabeled-Points $\leftarrow Embeddings[\sim \text{HighConfidence-}y]$
   Assuming size of Unlabeled-Points is $U$
5: Neighbors-Labels $\leftarrow kneighbors(\text{Unlabeled-Points}, k_p)$              $\triangleright \mathcal{O}(U \cdot log(U))$
6: Most-Common-Labels $\leftarrow mode(\text{Neighbors-Labels}, axis = 1)$             $\triangleright \mathcal{O}(U \cdot k_p)$
7: $Updated\text{-}\hat{y}[Updated\text{-}\hat{y} == -1] \leftarrow$ Most-Common-Labels
   To prevent converting a labeled point into an unlabeled again
8: Filter-Mask $\leftarrow \hat{y} \neq -1$ AND $Updated\text{-}\hat{y} == -1$
9: $Updated\text{-}\hat{y}[\text{Filter-Mask}] \leftarrow \hat{y}[\text{Filter-Mask}]$
10: **return** $Updated\text{-}\hat{y}$

---

## A.9    RESULTS WITH DIFFERENT EVALUATION METRICS AND WITH GROUND TRUTH $k$

Table 9 and Table 10 show the cluster performance results of DeepSynC and our competitors measured in normalised mutual information (NMI) (Vinh et al., 2010) and cluster accuracy (ACC) (Yang et al., 2010). If we rank the algorithms by counting how many of each algorithm achieved the best performance across different datasets, then from the ACC table, DeepSynC was the best, and DEC

Table 9: Average NMI (± std. deviation) on the 5 pretrained models. DS+= DeepSynC+, S=SynC, MS=MeanShift, AP=Affinity Propagation, DipD=DipDECK. [1] SynC was performed on 10% of the data due to full-run failures.

| Data Set | Ours | DS+ | AE+S | AE+MS | AE+AP | DEC | IDEC | DCN | DipD | ACe/DeC | DKM |
|----------|------|-----|------|-------|-------|-----|------|-----|------|---------|-----|
| **MNIST** | **81±01** | 80±01 | 67±03[1] | 00±00 | 47±00 | 75±05 | 78±01 | 73±02 | 62±03 | 69±01 | 72±02 |
| **FMNIST** | 61±01 | 58±01 | 52±03[1] | 00±00 | 41±00 | 62±02 | 63±02 | 59±01 | **64±01** | 59±01 | 61±04 |
| **USPS** | **80±02** | 75±02 | 44±15 | 00±00 | 54±00 | 79±02 | 78±01 | 73±03 | 79±02 | 62±02 | 76±01 |
| **Optdigits** | **89±01** | 87±01 | 79±01 | 00±00 | 60±00 | 82±03 | 79±03 | 77±03 | 78±06 | 72±04 | 76±01 |
| **COIL20** | **87±02** | 70±01 | 73±02 | 00±00 | 73±02 | 79±01 | 78±01 | 78±01 | 52±13 | 08±15 | 79±01 |
| **COIL100** | **94±01** | 79±00 | 33±09 | 00±00 | 81±01 | 86±01 | 85±00 | 84±01 | 15±03 | 00±00 | 84±01 |
| **Pendigits** | **88±01** | 72±02 | 61±04 | 00±00 | 60±00 | 77±03 | 74±01 | 74±03 | 81±01 | 72±02 | 74±04 |
| **HAR** | 60±03 | 60±02 | 59±06 | 55±00 | 44±00 | 72±02 | **74±05** | 72±06 | 72±00 | 62±02 | 73±00 |
| **HTRU** | 34±10 | 29±01 | 33±06 | 28±01 | 08±00 | 11±01 | 10±00 | 26±01 | 07±13 | 40±07 | **43±07** |
| **Letter** | **61±01** | 58±01 | 15±04 | 00±00 | 58±00 | 47±02 | 43±02 | 43±02 | 19±06 | 47±02 | 32±05 |
| **MICE** | 36±02 | 37±02 | 26±04 | 02±01 | **45±01** | 34±06 | 34±06 | 33±04 | 25±05 | 25±06 | 36±01 |
| **Weizm.** | 65±02 | 66±01 | 40±05 | 05±06 | 67±01 | **68±01** | 67±01 | 66±01 | 34±10 | 09±17 | 67±00 |

Table 10: Average Accuracy (± std. deviation) on the 5 pretrained models. DS+= DeepSynC+, S=SynC, MS=MeanShift, AP=Affinity Propagation, DipD=DipDECK. [1] SynC was performed on 10% of the data due to full-run failures.

| Data Set | Ours | DS+ | AE+S | AE+MS | AE+AP | DEC | IDEC | DCN | DipD | ACe/DeC | DKM |
|----------|------|-----|------|-------|-------|-----|------|-----|------|---------|-----|
| **MNIST** | **81±03** | 80±03 | 58±05 | 11±00 | 02±00 | 71±08 | 74±05 | 70±06 | 46±07 | 74±02 | 66±06 |
| **FMNIST** | 54±02 | 53±02 | 36±04 | 10±00 | 03±00 | 57±04 | **58±03** | 54±02 | 53±04 | 55±01 | 53±04 |
| **USPS** | **79±05** | 77±05 | 42±11 | 17±00 | 09±00 | 74±05 | 78±03 | 70±05 | 79±05 | 63±03 | 72±03 |
| **Optdigits** | **91±01** | 89±02 | 77±02 | 10±00 | 11±01 | 80±05 | 79±07 | 76±06 | 77±14 | 74±05 | 72±01 |
| **COIL20** | **77±03** | 50±03 | 51±04 | 05±00 | 37±04 | 66±02 | 67±02 | 68±03 | 38±15 | 10±10 | 67±02 |
| **COIL100** | **82±02** | 45±02 | 11±01 | 01±00 | 44±02 | 66±01 | 64±01 | 62±02 | 07±02 | 01±00 | 62±01 |
| **Pendigits** | **89±02** | 81±02 | 47±05 | 10±00 | 11±00 | 76±04 | 72±01 | 74±04 | 79±01 | 74±04 | 71±05 |
| **HAR** | 57±05 | 57±05 | 42±08 | 36±00 | 06±00 | 74±05 | **78±06** | 63±08 | 54±00 | 60±04 | 54±00 |
| **HTRU** | **96±00** | 92±01 | 92±01 | 90±01 | 01±00 | 59±01 | 59±00 | 92±00 | 91±01 | 93±01 | 94±01 |
| **Letter** | 34±02 | 30±01 | 10±02 | 04±00 | 09±00 | 35±01 | 31±01 | 32±02 | 12±02 | **36±02** | 20±03 |
| **MICE** | **41±02** | 40±01 | 28±02 | 15±01 | 18±01 | 36±04 | 37±05 | 37±04 | 27±05 | 32±04 | 37±02 |
| **Weizm.** | **40±02** | 34±02 | 12±02 | 03±01 | 31±02 | 38±02 | 37±03 | 37±01 | 15±03 | 06±06 | 37±01 |

==was second. However, according to the NMI table, DeepSynC was the best, and DeepSynC+ was second.==

==Further, we applied DeepSynC to all data sets, when we provided the ground truth $k$ to DeepSynC; the results are in Table 11. Comparing the results of DeepSynC and DeepSynC with given $k$, the results generally stay the same for most datasets. Results slightly improve on half of the data sets and get slightly worse for the other half. The pattern is similar for the results of DeepSynC+, where each unlabelled point is assigned to a singleton cluster.==

==Please also note that SHiP is a suitable choice for clustering the core points because it can automatically detect the number of clusters. However, if $k$ is provided, other algorithms may perform better and could consequently improve DeepSynC's performance. Additionally, it is a key advantage of DeepSynC is that it does not require users to know the number of classes in advance, fully adhering to the unsupervised learning paradigm.==

==In Table 11 we also include results for DeepSynC, when assigning all unlabelled data points to the nearest cluster. The results are shown in the column "Ours 1NN". Overall, the results do not differ much compared to DeepSynC+, although they are slightly worse in more of the data sets for Ours 1NN. On HTRU, on the other hand, results improve notably.==

Table 11: Average ARI of DeepSynC and DeepSynC+ with and without specifying the ground truth number of clusters. The last column has the average ARI after labeling all unlabeled points using 1NN.

| Data Set | Ours | DS+ | Ours True-K | DS+ True-k | Ours 1NN |
|---|---|---|---|---|---|
| **MNIST** | 75±02 | 74±02 | 77±05 | 76±05 | 73±02 |
| **FMNIST** | 40±02 | 39±02 | 38±01 | 37±01 | 39±02 |
| **USPS** | 75±05 | 72±04 | 76±04 | 73±03 | 70±05 |
| **Optdigits** | 88±02 | 86±02 | 81±06 | 80±06 | 85±03 |
| **COIL20** | 71±06 | 49±03 | 72±03 | 48±03 | 46±01 |
| **COIL100** | 79±06 | 53±02 | 85±01 | 53±02 | 46±02 |
| **Pendigits** | 84±02 | 72±01 | 81±03 | 70±01 | 68±03 |
| **HAR** | 46±05 | 46±05 | 45±07 | 45±07 | 45±05 |
| **HTRU** | 44±13 | 59±02 | 42±24 | 61±06 | 69±02 |
| **Letter** | 22±03 | 18±02 | 09±02 | 09±02 | 18±02 |
| **MICE** | 23±03 | 22±02 | 24±05 | 22±03 | 20±02 |
| **Weizm.** | 34±02 | 30±02 | 40±02 | 34±02 | 30±02 |

## A.10 THE SYNCHRONISATION MODEL BY KURAMOTO

On a very general notion, synchronisation can be described as the phenomenon of several units aligning their individual rhythms after interacting with each other. It has been studied in many scientific fields such as sociology, physics, biology and many others (Osipov et al., 2007). Mathematically, the interacting units are often modelled as so-called phase oscillators. These can be thought of as vectors moving along the unit circle. Their individual oscillatory movement is characterised by the frequency, with which they exercise their movement. Figure 13 illustrates this. The Kuramoto model is a model that describes the synchronisation of such oscillators mathematically. More specifically, it is a system of ordinary differential equations modelling how they adjust their phases $\varphi_i(t)$, $i = 1, \ldots r$ over time:

$$\frac{\partial \varphi_i(t)}{\partial t} = \frac{1}{r} \sum_{j=1}^{r} C \sin(\varphi_j(t) - \varphi_i(t)) \quad i = 1, \ldots, r. \tag{5}$$

Considering the phase oscillators as vectors moving around the unit circle, these phase values $\varphi_i(t)$ correspond to the deflection from the $y$-axis at time point $t$ (Figure 13). Each oscillator has its own natural rhythm, i.e. the individual movement around the unit circle, given by $\omega_i$, $i = 1, \ldots, r$, but interacts with the others through a coupling term, which is given by the sine of their phase differences. If the coupling strength set by the model parameter $C$ is strong enough, they shift from independent motion to a synchronised state. To assess the level of synchronisation at any given time point $t$, the *Kuramoto order parameter R(t)* can be calculated as:

$$R(t) = \frac{1}{r} \left| \sum_{k=1}^{r} e^{i\phi_k(t)} \right|. \tag{6}$$

Geometrically, this value corresponds to the length of the mean vector, when we describe each phase oscillator as a vector in polar coordinates $e^{i\phi_k(t)}$, such as seen in Figure 13.

To exploit the Kuramoto model for clustering the authors of the SynC algorithm Böhm et al. (2010) took the following approach. They interpret each data point as a phase oscillator and adapt the model to mimic the synchronisation of each coordinate of the data points. For that they derived the following iteration rule (step $t \to t + 1$) for each coordinate of every point in the data set:

$$x_i^{t+1} = x_i^t + \frac{1}{|\mathcal{U}_\varepsilon(x^t)|} \sum_{y^t \in \mathcal{U}_\varepsilon(x^t)} \sin(y_i^t - x_i^t), \tag{7}$$

where the model parameter $C$ is simply set to 1 and $\mathcal{U}_\varepsilon(x^t)$ is the $\varepsilon$- neighbourhood of $x$ in the data set $\mathcal{D}$ at iteration $t$:

$$\mathcal{U}_\varepsilon(x^t) = \{y^t \in \mathcal{D} | dist(x^t, y^t) < \varepsilon\}. \tag{8}$$

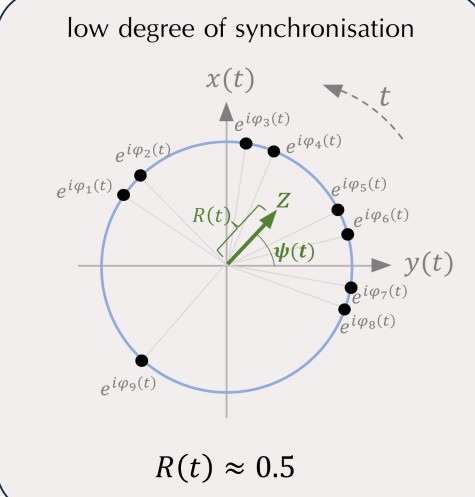
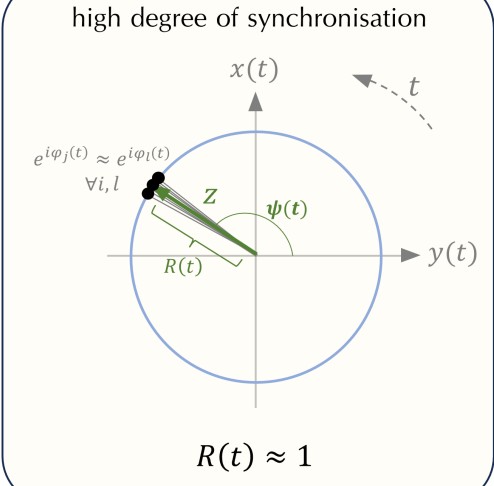

Figure 13: In the Kuramoto model the interacting units are viewed as oscillators, i.e. points moving counter-clockwise around the unit circle with their own individual rhythm, i.e. their own frequency. The current length of the average vector of the oscillators $R(t)$ is the Kuramoto Order Parameter (KOP). In case all individual phases $\varphi_k(t)$ are spread widely apart, the degree of synchronisation is low and $R(t)$ is far from its maximum of $1$ (left figure). In the case their phase values agree, i.e. $\varphi_j(t) \approx \varphi_l(t)$, $R(t)$ approches its maximum of $1$ and the degree of synchronisation is high.

Although it is not a part of their clustering methodology, they also adopted the KOP for the context of clustering:

$$r^t := \frac{1}{N} \sum_{i=1}^{N} \frac{1}{|\mathcal{U}_\varepsilon(x^t)|} \sum_{y^t \in \mathcal{U}_\varepsilon(x^t)} e^{-||y^t - x^t||_2^2}. \qquad (9)$$

It measures the strength of the synchronisation between data points at time point $t$ within a certain $\varepsilon$-neighbourhood $\mathcal{U}_\varepsilon$. This parameter lies between 0 and 1, where 1 indicates a perfect (local) synchronisation. This adaptation of the KOP gave us the inspiration for formationg a synchronisation-based cluster loss. Instead of simulating the synchronisation behaviour based on the initial positions of the data points and observing the state of synchronisation with the KOP, we developed a new cluster loss function $\mathcal{L}_{clu}$ based on the KOP, such that the autoencoder aims at synchronisation.

### A.11 EFFECTS OF HYPERPARAMETERS '$|\mathcal{B}|$' AND '$m$'

In two experiments, we investigated the effects of the two hyperparameters batch size $|\mathcal{B}|$ and embedded space dimension $m$ on the performance of DeepSynC and its competitors.

For testing the effect of varying batch sizes on the cluster losses of DeepSynC and its competitors, we pre-trained five models of each method for each of the four data sets MICE, Optidigits, HAR, and USPS with the same setting as in Section 4. Hence, we kept the pre-training batch size fixed as 256, however, we reduced the number of pre-training epochs to 30 to allow faster computations. As the main goal was to investigate the effect of the batch size on the specific losses

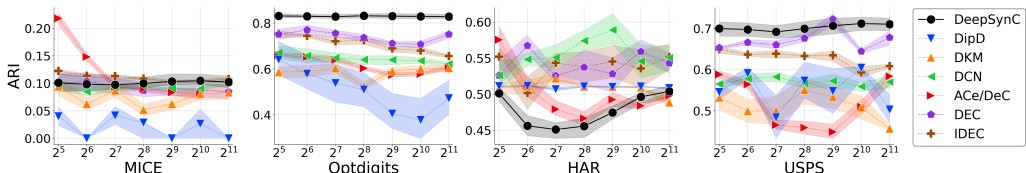

Figure 14: For each of the four data sets MICE, Optdigits, HAR and USPS, we observed how the performance is affected by varying batch sizes ranging from $2^5$ to $2^{11}$. All methods are affected, but overall DeepSynC shows more stability.

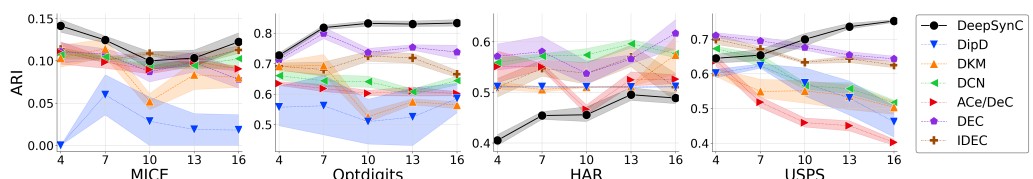

Figure 15: For each of the four data sets MICE, Optdigits, HAR and USPS, we observed how the performance is affected by varying embedded space dimensions ranging from 4 to 16. All methods are affected, but overall, DeepSynC shows as much stability as competitors.

of DeepSynC and its competitors, which are only relevant in the clustering epochs, this does not bias the results. In the cluster training epochs, each model was trained with each batch size within $\{2^5, 2^6, 2^7, 2^8, 2^9, 2^{10}, 2^{11}\}$. The results can be seen in Figure 14. The lines with the markers show the average ARI values across the 5 models, and the error bars are the 50% confidence intervals for better visualisation. As expected, all algorithms are to a certain extent affected by varying batch sizes. However, DeepSync shows more stability in its performance in comparison. Hence, there is no evidence that our pairwise synchronisation-based loss is more sensitive to the batch size than the loss of competitors.

For the experiments with varying embedded space dimensions, we trained five autoencoder models for each of the four data sets MICE, Optidigits, HAR, and USPS and each embedded space dimension within $\{4, 7, 10, 13, 16\}$. Apart from the embedded space dimension, we also set the number of pre-training epochs to 30 to allow faster computation. All other settings are kept as in Section 4, including a fixed batch size of 256 for pre-training and cluster training. The results can be seen in Figure 15. The lines with the markers show the average ARI values across the 5 models, and the error bars are the 50% confidence intervals for better visualisation. As for varying batch sizes, all algorithms are to a certain extent also affected by varying embedded space dimensions. DeepSynC shows at least as much stability as the competitors in this regard.

