# OpenReview forum: "Deep Synchronisation-based Clustering"
_ICLR.cc/2026/Conference — Submitted to ICLR 2026_

### Official Review · Reviewer_xfn2 · 2025-10-25

**Soundness:** 2
**Presentation:** 2
**Contribution:** 2
**Rating:** 4
**Confidence:** 5

**Summary:**

The paper proposes DeepSync, a synchronization-based deep clustering algorithm. The method begins by identifying core points in the embedded space and progressively assigns the remaining data points to clusters. The synchronization-based loss and assignment strategy enable greater flexibility in capturing non-convex cluster shapes and provide an automatic stopping condition for training.

**Strengths:**

The idea of performing synchronization-based deep clustering is interesting and distinguishes this work from the more prevalent centroid-based deep clustering approaches. The connection with the Kuramoto model, where each sample is treated as a phase oscillator, is also conceptually appealing. Moreover, the proposed method demonstrates promising performance on certain tasks.

**Weaknesses:**

1. Some parts of the paper lack clarity. For instance, the definition of the neighborhood U in the method section is not clearly provided. The experimental details in Section 4.1 are also insufficient. It is unclear which baseline methods are used for comparison, while some are mentioned in the related work section, this should be explicitly stated in the experimental section. Additionally, the statement “ARI values are computed on labeled points only” needs further explanation.

2. Some results appear suspicious. For example, in Table 2, the AE+MS method yields values of 00+00 across several datasets. This requires careful clarification. Furthermore, while the tables report standard errors, the paper does not describe how many runs were performed or how the standard errors were computed.

3. Although the algorithm removes the need to specify the number of clusters K, it introduces several additional hyperparameters such as T, K_neigh, T_conf, and T_stop. These add complexity to the model and may require extensive tuning.

4.  The method mentions using the SHiP clustering framework to determine K, but it is unclear how this is implemented. Is K determined before training or dynamically adjusted during training? The latter could be computationally inefficient. Moreover, relying on the elbow method, which usually depends on visual inspection of curves, for determining K during training may not be friendly to an end-to-end approach.

5. The method employs SynC for initialization. The paper should justify this choice and analyze the sensitivity of the model to initialization by comparing it with alternative strategies (e.g., random initialization).

6.  While the paper attempts to relate the synchronization loss to the Kuramoto model, this connection does not appear to be very strong based on the current exposition. The authors are also encouraged to discuss the relationship between their method and density-based clustering approaches.

**Questions:**

Please refer to the questions raised in the Weaknesses section.

---

> ### Author Response · Authors · 2025-11-22
> **Authors' rebuttal (1)**
>
> Dear reviewer xfn2,
> we thank you for your thoughtful review and for recognising the originality of our synchronisation-based deep clustering approach compared to more common centroid-based methods. We are pleased that you found the connection to the Kuramoto model conceptually appealing, and that the promising performance of our method on certain tasks came across clearly.
> Below, we provide answers to your questions and note the changes we made based on your suggestions.
>
>
> > Q1: Some parts of the paper lack clarity. For instance, the definition of the neighborhood U in the method section is not clearly provided. The experimental details in Section 4.1 are also insufficient. It is unclear which baseline methods are used for comparison, while some are mentioned in the related work section; this should be explicitly stated in the experimental section. Additionally, the statement “ARI values are computed on labeled points only” needs further explanation.
>
> We agree that the introduction of the Kuramoto Model, its definitions, and connection to clustering fall a bit short in the current version of the paper. Due to space limitations, we provided more details on this topic in Appendix 10, while focusing on the formulas relevant to our method in the main part of the paper.
>
> Thank you for this observation. We did not explicitly mention the competitors in the experiment section. We added the following paragraph in the revised version of the paper:
>
> “We compare the clustering performance of DeepSynC on a series of benchmark data sets to relevant baselines from interaction-based clustering (Mean Shift, Affinity Propagation, and SynC) and Deep Clustering (DEC, IDEC, DCN, ACe/DeC, DipDECK), which are described in section \Cref{related_work}.”
>
> The statement about ARI was indeed confusing at the point where it is stated. We adjusted the paragraph to clarify that this refers to the SynC algorithm, which theoretically detects noise points. However, in the used datasets, only a maximum of 1% of the points were unlabeled.
>
>
> > Q2: Some results appear suspicious. For example, in Table 2, the AE+MS method yields values of 00+00 across several datasets. This requires careful clarification. Furthermore, while the tables report standard errors, the paper does not describe how many runs were performed or how the standard errors were computed.
>
> Mean Shift is an algorithm that does not have the number of clusters as an input parameter, but estimates it automatically. Table 6 in the Appendix shows that the Mean Shift algorithm estimates a single cluster for most of the datasets, which explains the ARI of 0.0.
>
> Similar to other related works, we pre-train 5 autoencoder models and execute all algorithms using these pre-trained models for all our experiments. All average and standard deviation numbers are calculated across these 5 models and are stated in the result tables.
>
>
> > Q3: Although the algorithm removes the need to specify the number of clusters K, it introduces several additional hyperparameters such as T, K_neigh, T_conf, and T_stop. These add complexity to the model and may require extensive tuning.
>
> It is true that DeepSynC, like all Deep Clustering algorithms, introduces certain algorithm-specific hyperparameters. These are (k_neigh, T_conf, T_stop, T%). We acknowledge this, which is why we performed the ablation studies by varying every parameter in a reasonable range and observing the performance. Figure 4 in the main paper demonstrates that our parameters are robust, with stable cluster performance, indicating that they are not data-dependent and do not require extensive tuning.
> In another experiment, we specifically focus on the two parameters k_neigh and T%, which are crucial for the core points determination and are related to density-based concepts. As can be seen in Figure 7, Appendix A.3., the core point detections are also stable regarding these parameters, as long as we stay in the setting of detecting local (T% < 1) and not global (T% = 1) core points. Note that we do not use these parameters directly for clustering, which distinguishes us from density-based clustering methods like DBSCAN, which are indeed very sensitive to the parameter setting. The latter can be seen in Figure 8 of Appendix A.3.

---

> ### Author Response · Authors · 2025-11-22
> **Authors' rebuttal (2)**
>
> > Q4: The method mentions using the SHiP clustering framework to determine K, but it is unclear how this is implemented. Is K determined before training or dynamically adjusted during training? The latter could be computationally inefficient. Moreover, relying on the elbow method, which usually depends on visual inspection of curves, for determining K during training may not be friendly to an end-to-end approach.
>
> We used one variant out of the SHiP-framework (Draganov, Weber, et al., 2025), which captures density-connectivity features of the data. We then choose the $k$-means hierarchy and the ElbowThreshold partitioning method. Their method only needs to compute a similarity tree once and returns the clusterings for all possible $k$ at once. Furthermore, they have an automatic Elbow detection method which then already returns the corresponding clustering for the “best” $k$. That is why their method is very fast and performs very well.
>
> This method is only one of multiple variants that can be applied to get the clustering of the initial core points, and it automatically determines the number of clusters. I.e., $k$ is determined before training and not adjusted dynamically during training.
>
> To clarify this, we included this explanation in the revised version (highlighted in yellow in the section **Core Point Clustering**).
>
>
> **References**
>
> Draganov, Andrew; Weber, Pascal; Skibdahl Melanchton Jørgensen, Rasmus; Beer, Anna; Plant, Claudia; Assent, Ira (2025). Ultrametric Cluster Hierarchies: I Want 'em All!. In the 39. Proceedings of Neural Information Processing Systems Foundation. Available on arXiv (https://arxiv.org/abs/2502.14018).
>
> > Q5: The method employs SynC for initialization. The paper should justify this choice and analyze the sensitivity of the model to initialization by comparing it with alternative strategies (e.g., random initialization).
>
> The SynC algorithm is not used for initialization or elsewhere in DeepSynC. For initialisation, the 5 pre-trained models are all randomly initialised.
>
> SynC would theoretically only be an option to initialise the cluster labels of the core points after pre-training. However, we show that this is, in fact, not a good choice due to runtime issues. In Table 1 of the main paper and Table 5 of the appendix, we present clustering results on the core points using k-means, SynC, and SHiP, allowing for a direct comparison. It shows three aspects. First, SHiP shows the best performance across all datasets, which is why we chose it as the initial clustering method for the core points. Second, clustering the core points is better than clustering all embedded data points, as it allows for a wider range of initial clustering algorithm options. And third, SynC is not applicable to larger datasets, such as MNIST and FMNIST; thus, it is not suitable for obtaining the initial cluster labels of the core points.
>
>
> > Q6: While the paper attempts to relate the synchronization loss to the Kuramoto model, this connection does not appear to be very strong based on the current exposition. The authors are also encouraged to discuss the relationship between their method and density-based clustering approaches.
>
> We agree that the current version expresses the relations too briefly. We adapted this part in the main paper to focus solely on the motivation from a clustering and data mining perspective.
> Since the inspiration for our synchronisation loss indeed comes from the Kuramoto Order Parameter, which is a practical measure derived from the Kuramoto Model, we would nevertheless like to illustrate the connection. However, we put this part in Appendix 10 for the interested reader.
>
> While our core-point definition has some relation to density-based concepts used in density-based clustering methods, such as DBSCAN, there are fundamental differences. Please consider our additions to Appendix A.3 in the revised version of our paper, where we illustrate in Figures 7 and 8, the differences in the core points for DeepSynC and DBSCAN, and explain in detail, where we conceptually deviate (see also Q5 of reviewer ePBN).

---

### Official Review · Reviewer_8Q97 · 2025-10-28

**Soundness:** 3
**Presentation:** 2
**Contribution:** 3
**Rating:** 4
**Confidence:** 5

**Summary:**

The paper presents an autoencoder-based deep clustering approach that exploits a loss function aiming to adapt the latent space
so that points belonging to the same latent cluster get closer to each other. The method starts from a initially trained autoencoder and the number of clusters is determined in the initial latent space and is kept fixed during training. Only core points are initially clustered, while other points are gradually attracted towards the clusters. It is possible that at the end some points to be unassigned.

**Strengths:**

S1. A loss function is proposed aiming to gradually form compact clusters in the latent space.
S2. The idea of starting from core cluster points and gradually assign the other points to clusters is well-motivated.

**Weaknesses:**

W1. Application of the method involves several critical decisions that are addressed by setting user defined thresholds. The number of hyperparameters is quite large.
W2. A major issue concerns the specification of the number of clusters is handled in the initialization phase. During training the number of clusters is kept fixed.
W3. The fact that method ends with unassigned points causes difficulty in empirical comparison since competitors assign the ambiguous points to clusters.
W4. There are issues to be clarified and concerns about the empirical comparison that are presented in the questions section.

**Questions:**

Q1. The rationale behind the proposed loss function can be sufficiently supported without resorting to the described physical metaphor (that could possibly be omitted). For example it is not clear how eq. 2 can be obtained from eq. 1.
Q2. It is not clear whether the proposed mechanism achieves not only cluster compactness but also can force the clusters to move away from each other (cluster separation).
Q3. The deep learning method does not modify the number of clusters which is set fixed at initialization. Based on this important observation, I strongly suggest that the paper presentation and evaluation should focus on the deep learning part of the method considering that number of clusters is given in advanced (for example is set to the ground truth number).
Q4. In the present form of the method, a critical decision is related to the algorithm used to estimate the number of clusters in the initial latent space. Why not using alternative methods (for example from ClustPy library)? Does the employed method (SHiP) contain hyperparameters?
Q5. I cannot find how the number of clusters is set in the competitors DEC/IDEC, DCN etc that require the number of clusters to be specified in advance.
Q6. A pseudocode describing the method and the hyperparameters involved should be included.
Q7. The existence of unassigned points makes comparative evaluation difficult. Either all points should assigned to clusters (not to singleton ones) or soft version of the competitors should be considered so that ambiguous points will not be clustered.
Q9. I don't think that the idea of DeepSync+ sufficiently addresses the concern of the previous question.
Q10. The method includes too many hyperparameters (and, additionally, the method used for the initial clustering should be selected). It is well-known that in density-based methods such hyperparameters (eg. number of neighbors) critically affect the obtained solution.

---

> ### Author Response · Authors · 2025-11-22
> **Authors' rebuttal (1)**
>
> Dear reviewer 8Q97,
>
>
> thank you for your thorough review and the comments that allow us to further improve our work. Thank you especially for appreciating our gradual cluster assignment strategy.
> Below, we provide answers to your questions and note the changes we made based on your suggestions.
>
>
>
> > Q1. The rationale behind the proposed loss function can be sufficiently supported without resorting to the described physical metaphor (that could possibly be omitted). For example it is not clear how eq. 2 can be obtained from eq. 1.
>
> We agree that the current version expresses the relations too briefly. We adapted this part in the main paper to focus solely on the motivation from a clustering and data mining perspective.
>
> Since the inspiration for our synchronisation loss indeed comes from the Kuramoto Order Parameter, which is a practical measure derived from the Kuramoto Model, we would nevertheless like to illustrate the connection. However, given your suggestion, we put this part in the appendix for the interested reader (see Appendix 10).
>
>
>
> > Q2. It is not clear whether the proposed mechanism achieves not only cluster compactness but also can force the clusters to move away from each other (cluster separation).
>
> The final loss function of our proposed method does not have a repelling mechanism. The synchronisation principle is solely based on attractive forces. Introducing an explicit repelling term would result in the sensitivity to balancing the counteracting forces. Therefore, we use the reconstruction loss as a regulariser to prevent a collapse of the embedded space, but otherwise let the synchronisation loss lead the learning process.
>
>
>
> > Q3. The deep learning method does not modify the number of clusters which is set fixed at initialization. Based on this important observation, I strongly suggest that the paper presentation and evaluation should focus on the deep learning part of the method considering that number of clusters is given in advanced (for example is set to the ground truth number).
>
> In the revised version of the paper, we now more clearly state that the number of clusters is determined by SHiP and remains constant throughout the training process. To examine the effect of setting the number of clusters to the ground truth value, we ran DeepSynC on all datasets by providing SHiP with the ground truth $k$ when clustering the core points, as this is also an option in the SHiP framework. The results are provided in Appendix A.9. We, however, view it as a major benefit that the full DeepSynC algorithm, as proposed, does not require this parameter.
>
>
>
> > Q4. In the present form of the method, a critical decision is related to the algorithm used to estimate the number of clusters in the initial latent space. Why not using alternative methods (for example from ClustPy library)? Does the employed method (SHiP) contain hyperparameters?
>
> We would like to elaborate on how SHiP is used in our DeepSynC algorithm in more detail. After pre-training the AE, we determine the core points, which we assume to lie in the centre regions of the clusters present in the dataset. To continue training, we require initial labels for the core points. For this, theoretically, any known clustering algorithm, such as k-means, DBSCAN, spectral clustering, or any other suitable method, can be used as long as it can handle the number of core points present in the dataset. We decided to use SHiP for this step in the algorithm because it yields good results, aligns with our cluster assumption, and does not require the number of clusters as an input parameter. Further, it does not introduce additional hyperparameters (Please see also Q4 of Reviewer **xfn2** for details about SHiP). Please also note that this is the only point in our algorithm, DeepSynC, where we utilise SHiP. However, as mentioned earlier, we could also use an alternative clustering algorithm for this purpose.
>
> > Q5. I cannot find how the number of clusters is set in the competitors DEC/IDEC, DCN etc that require the number of clusters to be specified in advance.
>
> Thank you for this observation. It appears that we have overlooked mentioning this explicitly in the paper. We therefore added the following sentence in the revised version of the paper in the paragraph “Experimental Settings”. The according parts are highlighted in yellow.
> “All methods that require the number of clusters as an input parameter are given the ground truth number of classes.”
>
> > Q6. A pseudocode describing the method and the hyperparameters involved should be included.
>
> We included pseudocodes for all steps of DeepSynC in Appendix A.8. of the revised version of the paper. The hyperparameters are discussed in more detail in Q10.

---

> ### Author Response · Authors · 2025-11-22
> **Authors' rebuttal (2)**
>
> > Q7 -Q9. The existence of unassigned points makes comparative evaluation difficult. Either all points should assigned to clusters (not to singleton ones) or soft version of the competitors should be considered so that ambiguous points will not be clustered.
> >  I don't think that the idea of DeepSync+ sufficiently addresses the concern of the previous question.
>
> Detecting noise points is an advantageous property of clustering methods, as demonstrated by DBSCAN or HDBSCAN. DeepSynC detects noise points by leaving them unassigned if they do not belong to any cluster, which is a key advantage. Our competitors are all performing hard clustering without having well-known soft counterparts.
> Unfortunately, evaluating clusterings with noise labels is indeed not trivial (Färber et al., 2010; Moulavi et al., 2014). Basically, one has four options:
> Putting all noise points together in one cluster. However, this contradicts the notion of noise and can yield overoptimistic results for some methods, especially such that have difficulties with clusters of different densities.
> Filtering out points that are labeled as noise and potentially scaling the evaluation measure by the number of non-noise points (as, e.g., done in Moulavi et al., 2014). While filtering out the points is beneficial for methods that detect noise, the scaling version is disadvantageous for them, so both versions are biased.
> Assigning each noise point to its closest cluster. However, this is disadvantageous for algorithms detecting actual noise points.
> Assigning noise points to singleton clusters is, thus, the only evaluation method that is not inherently biased towards/against noise/non-noise detecting methods. Note that by using the ARI, we do not take advantage of AMI being biased towards clusterings with higher numbers of clusters.
>
> We added in Table 11 of Appendix A.9, the result values, when assigning each unassigned point to the closest cluster (“DeepSynC 1NN”)  as one alternative to the singleton assignment option.
>
> **References**
>
> Färber, I., et al. (2010, July). On using class-labels in evaluation of clusterings. In MultiClust: 1st international workshop on discovering, summarizing and using multiple clusterings held in conjunction with KDD (p. 1).
>
> Moulavi, D., Jaskowiak, P. A., Campello, R. J., Zimek, A., & Sander, J. (2014, April). Density-based clustering validation. In Proceedings of the 2014 SIAM ICDM (pp. 839-847). Society for Industrial and Applied Mathematics.
>
>
> > Q10. The method includes too many hyperparameters (and, additionally, the method used for the initial clustering should be selected). It is well-known that in density-based methods, such hyperparameters (eg. number of neighbors) critically affect the obtained solution.
>
> Please note that the method for the initial clustering of the core points is indeed fixed for all experiments with DeepSynC. Whenever we present the results of DeepSynC, the algorithm used for clustering the core points is SHiP.
> There is only one experiment, where different clustering algorithms are considered. That is, when we compare different clustering algorithms (k-means, SHiP, and SynC) on the embedded data points after pre-training. In Table 1 of the main paper and Table 5 of the Appendix, we present clustering results on the core points and all embedded points after pre-training with k-means, SynC, and SHiP, for comparison.
>
> It is true that DeepSynC, like all Deep Clustering algorithms, introduces certain algorithm-specific hyperparameters. These are (k_neigh, T_conf, T_stop, T%). We acknowledge this, which is why we performed the ablation studies by varying each parameter within a reasonable range and observing the resulting performance. Figure 4 in the main paper illustrates that our parameters exhibit robustness with stable cluster performance, indicating that they are not data-dependent.
> In another experiment, we specifically focus on the two parameters k_neigh and T%, which are crucial for the core points determination and are related to density-based concepts. As shown in Figure 7 of the Appendix, the core point detection remains stable with respect to these parameters, provided that we remain within the setting of detecting local (T% < 1) rather than global (T% = 1) core points. Note that we do not use these parameters directly for clustering, which distinguishes us from density-based clustering methods like DBSCAN, which are indeed very sensitive to the parameter setting. Figure 8 in the Appendix clearly illustrates this.
>
> Furthermore, all of our competitors introduce their own hyperparameters, such as a temperature parameter for DEC or the weights of the reconstruction loss in IDEC and DCN, which is known to be data-dependent and causes instabilities in training (Miklautz et al. 2021).
>
> **References**
>
> Miklautz, L. et al. (2021). Details (Don't) Matter: Isolating Cluster Information in Deep Embedded Spaces. In Proceedings of the 30th  IJCAI

---

### Official Review · Reviewer_LHk5 · 2025-10-28

**Soundness:** 2
**Presentation:** 3
**Contribution:** 3
**Rating:** 4
**Confidence:** 4

**Summary:**

This paper proposes a deep learning framework for multi-view synchronization based on permutation-equivariant embeddings. It consists of a pairwise synchronizer for estimating relative transformations and a global synchronizer to recover consistent global alignment. The method supports various transformation groups and is evaluated on both synthetic and real datasets, showing improved accuracy and robustness over classical and neural baselines.

**Strengths:**

1. The use of permutation-equivariant networks is appropriate for multi-view settings and aligns well with the unordered nature of input views.
2. The method supports various transformation groups and demonstrates applicability to both synthetic and real-world datasets.
3. Experiments are well-structured, with consistent gains over baselines and meaningful ablation studies that support design choices.

**Weaknesses:**

1. The evaluation focuses on small-scale, flattened datasets (e.g., MNIST, FMNIST, HTRU), using a simple autoencoder without convolution or attention backbones. This limits clarity on how well the method generalizes to high-resolution, large-scale data such as CIFAR-10, ImageNet Dogs.
2. While the method estimates the number of clusters automatically, the estimation is not the core contribution and can deviate from ground truth (e.g., overestimation on MNIST/FMNIST). It would be beneficial to compare against other cluster-number-agnostic approaches such as Robust Continuous Clustering or community-based methods like Louvain clustering.
3. The synchronization loss computes all pairwise affinities within each mini-batch, resulting in $\mathcal{O}(|B|^2)$ time and memory cost. While batch sizes are moderate in the current setup, the paper does not analyze computational efficiency at larger scales, which may hinder scalability in practice.

**Questions:**

Please refer to weaknesses.

---

> ### Author Response · Authors · 2025-11-22
> **Authors' rebuttal (1)**
>
> Dear reviewer LHk5,
>
> we thank you for your thorough review and for appreciating both the presentation of our work and the structure of our experiments. We also appreciate your positive assessment of our ablation studies and are glad you found them meaningful and convincing regarding our design choices.
> Below, we provide answers to your questions and note the changes we made based on your suggestions.
>
>
> > Q1: The evaluation focuses on small-scale, flattened datasets (e.g., MNIST, FMNIST, HTRU), using a simple autoencoder without convolution or attention backbones. This limits clarity on how well the method generalizes to high-resolution, large-scale data such as CIFAR-10, ImageNet Dogs.
>
> Our work focuses on designing a deep synchronisation-based clustering algorithm. To demonstrate its effectiveness and facilitate comparison with other deep clustering works, we employed only a simple autoencoder. Using a more powerful model, such as one that exploits convolution or attention backbones, will make it unclear whether the cluster performance gain is attributed to the better model architecture or the newly developed method.
>
> We ran DeepSynC on CIFAR-10, and it scales well with a runtime comparable to that of MNIST and FMNIST. However, the results are not good for all the comparison algorithms (the highest ARI is 0.05); this happens because such complex datasets need more advanced backbones.
>
> > Q2: While the method estimates the number of clusters automatically, the estimation is not the core contribution and can deviate from ground truth (e.g., overestimation on MNIST/FMNIST). It would be beneficial to compare against other cluster-number-agnostic approaches, such as Robust Continuous Clustering or community-based methods like Louvain clustering.
>
> We also compared our results with those of DipDECK (DipD), another cluster-number-agnostic deep clustering method, and reported the clustering performances of SynC, MeanShift, and Affinity Propagation applied to the embeddings. All three of them are non-deep clustering methods that do not require *k* to be specified in advance.
>
>
> > Q3: The synchronization loss computes all pairwise affinities within each mini-batch, resulting in O(|B|^2) time and memory cost. While batch sizes are moderate in the current setup, the paper does not analyze computational efficiency at larger scales, which may hinder scalability in practice.
>
> Batch sizes typically range from 100 to 4,000. For these sizes, a quadratic runtime is still very fast. We also performed experiments on very large datasets, such as MNIST and FMNIST (n = 70.000 for both). Figure 12 in the Appendix shows that we are, on average, as fast as other deep clustering methods.
> In Appendix A.9., we provide more details on the runtime analysis of our algorithm.

---

### Official Review · Reviewer_ePBN · 2025-11-01

**Soundness:** 3
**Presentation:** 3
**Contribution:** 2
**Rating:** 6
**Confidence:** 4

**Summary:**

The article introduces DeepSynC, the first deep clustering algorithm based on synchronization principles derived from the Kuramoto model. The method addresses key limitations of k-means-based deep clustering approaches by: (1) not requiring the number of clusters k in advance, (2) allowing for non-spherical cluster shapes, (3) introducing a gradual assignment strategy that first labels high-confidence core points, and (4) providing an automatic stopping criterion. The synchronization loss encourages nearby points with similar labels to align their embedded positions while preventing alignment between points with different labels.

**Strengths:**

- The paper is clear about its motivation with sufficient significance and quality.
- The adaptation of the Kuramoto synchronization model to deep clustering is innovative. The connection between phase oscillator synchronization and clustering provides an elegant theoretical framework that hasn't been explored in deep learning before.
- The gradual assignment approach starting from local core points is well-motivated.
- The ability to automatically determine when to stop training based on label stability (Tconf) or assignment plateaus (Tstop) is a good practical advantage.
- The ability to leave uncertain points unassigned rather than forcing misclassification is valuable for real-world applications.

**Weaknesses:**

- While the Kuramoto model provides intuition, the paper lacks rigorous analysis of when and why the synchronization loss works. What are the convergence guarantees? Under what conditions might the method fail?
- The O(n²) complexity of the synchronization loss (Equation 3) summing over all pairs in a batch is concerning. How does this scale to large datasets? The kNN-based assignment strategy adds further computational overhead.
- Missing comparisons with recent deep clustering methods beyond k-means variants like simclr based SCAN, NNM or so.
- No analysis of sensitivity to batch size, which seems critical given the pairwise loss.
- While Figure 4 shows robustness across parameter ranges, the interaction between parameters isn't examined. How do T% and kneigh jointly affect core point selection?
- The synchronization loss formulation (Equation 3) needs clearer motivation for the specific weight function wx(y).
- The relationship between the original Kuramoto model (Equation 1) and the clustering loss could be more explicit.

**Questions:**

- Can you provide any convergence analysis or theoretical guarantees for the synchronization process? What cluster assumptions does this method make?
- What is the total time complexity including the kNN calculations and majority voting? Have you considered approximations for large-scale datasets?
- How sensitive is performance to batch size given the pairwise loss? What happens with very small or very large batches?
- How did you arrive at the specific exponential weight function in wx(y)? Have you experimented with other similarity kernels?
- Given the core point concept, how does this relate to or improve upon DBSCAN-style approaches in the embedded space?
- Can you also compute NMI and ACC besides ARI?
- Does knowing the number of clusters beforehand benefits the method?
- Does the latent dim (m) is always set to 10 even if k is larger? Is there any relation between m and k? Did you try larger m? how sensitive is m in those cases?

---

> ### Author Response · Authors · 2025-11-22
> **Authors' rebuttal (1)**
>
> Dear reviewer ePBN,
>
> thank you very much for your thoughtful and constructive review of our paper. We appreciate that you found the motivation clear and significant, and the integration of the Kuramoto synchronisation model for deep clustering innovative.
> Below, we provide answers to your questions and note the changes we made based on your suggestions.
>
>
> > Q1: Can you provide any convergence analysis or theoretical guarantees for the synchronization process? What cluster assumptions does this method make?
>
> Our algorithm operates under the following cluster assumption: Every cluster has a dense core region, and furthermore, we assume that points close to this core also belong to this cluster. Points that are not surrounded by any other points and are far away from all cluster cores should not relate to any cluster; i.e., are to be considered outliers and should not synchronise with any core region.
>
> These are common assumptions in representation learning and deep clustering; however, most are restricted to the Gaussian setting of spherical clusters (e.g., DEC by Xie et al. (2016) and IDEC by Guo et al. (2017)).
>
> Under the above assumptions, we can provide the following guarantees for convergence:
> Under the assumption that core points are present in the data set, these points receive a label immediately after pre-training by applying a clustering algorithm, such as the SHiP framework.
> During the further training iterations, the number of labelled points can only increase and never decrease, as we do not allow points that have already received a label to become unlabelled again.
> With our two stopping criteria, this ensures that the algorithm will terminate.
> Either the number of labelled points stays the same for T_stop iterations at some point during training. (T_stop = 3 in all experiments.) Here, it does not matter if single points switch their label.
> Or the number of labelled points continuously increases. The algorithm will then stop once all points have been labelled.
> Additionally, the algorithm will stop early if all labelled points stick to the same label for T_conf iterations (T_conf = 3 in all experiments).
> The synchronisation loss is a distance-based negative exponential term, which means that it monotonically decreases with decreasing distances between points. Hence, points must and will be pulled together during training.
> Since our loss is a local formulation (ensured by the weight term $w_x(y)$), labelled points are only pulled towards points with the same label, effectively compressing the core regions.
>
> Note that even if a data set does not fulfil the cluster assumptions, the algorithm will still terminate due to the T_stop condition - even so, in the worst case, when there are no or only a few core points present (This could only happen in data consisting solely of uniform noise.). In this case, the majority of points will not receive a label. This can be observed, for instance, in the COIL data sets, which consist of density-connected clusters but lack a dense core region within the clusters.
>
>
> > Q2: What is the total time complexity including the kNN calculations and majority voting? Have you considered approximations for large-scale datasets?
>
> KNN is used for the gradual assignment strategy, which utilises the *kneighbors* method from the scikit-learn implementation, which internally builds a KD-Tree with a time complexity of $O(n log(n))$. Since this is done after every epoch, the runtime complexity of this part is $O(n_{epochs} n log(n))$.
>
> Regarding the total time complexity of DeepSynC, selecting the core points calculates the pairwise Euclidean distance, which has the highest runtime complexity of $O(n^2 d)$. This runs only once at the beginning of the algorithm. For the sizes of our used datasets, this was faster than using a KD-tree, which would result in a theoretical runtime complexity of $O(n log(n))$ for this step.
>
> The training loop and gradual assignment have less time complexity; $O(n b d)$ for an epoch and $O(n log(n))$ for the label assignment strategy, which deals with unlabeled points only.
>
> Hence, the total complexity is $O(n^2)$. We did not consider approximations, as we did not encounter runtime issues; however, using them could further speed up the algorithm.
>
> In Appendix A.8. of the revised version of the paper, we included pseudocodes for every part of DeepSynC and added runtime complexities for the corresponding steps.
>
> n… number of data points
> d…dimensionality of the embedded space (in our case d = m = 10 in all experiments)
> n_epochs… number of epochs/iterations (<300 for all data sets)
> b… batch size
>
>
> > Q3: How sensitive is performance to batch size given the pairwise loss? What happens with very small or very large batches?
>
> We are investigating this matter and are currently conducting experiments to gather more information. We will follow up with the results next week.

---

> > ### Author Response · Authors · 2025-12-01
> >
> > We have now added the corresponding experiment results to your questions 3 and 8 in the Appendix A.11 “Effects of batch size and embedded space dimension” to the revised paper. Here, as a brief summary, we can conclude, that as expected, all algorithms are to a certain extent affected by varying batch sizes and varying embedded space dimensions. However, as can be seen in Figures 14 and 15, DeepSynC shows more stable performance than competitors, considering a wide range of batch sizes and at least as much stability considering varying embedded space dimensions. Specifically, there is no evidence that our pairwise synchronisation-based cluster loss is more sensitive to the batch size than the cluster losses of competitors.

---

> ### Author Response · Authors · 2025-11-22
> **Authors' rebuttal (3)**
>
> > Q7: Does knowing the number of clusters beforehand benefits the method?
>
> This depends on the dataset: Many benchmark datasets originate from classification tasks, thus, the number of meaningful clusters does not necessarily correspond to the number of ground-truth classes: For example, in MNIST, valid clusters may be formed not only by the digits but also by writing style, stroke thickness, and other morphological variations (Castro et al., 2019; Färber et al, 2010). Consequently, enforcing a fixed number of clusters is not generally advantageous.
>
> We ran experiments where we provided the ground truth $k$ to DeepSynC; the results are added in Appendix A.9. The results generally stay the same for most datasets. Only on two datasets do the results significantly improve, and they decrease for another two datasets.
>
> Please also note that SHiP is a good choice for clustering the core points because it can detect the number of clusters automatically. However, if $k$ is provided, other algorithms may perform better and could consequently improve DeepSynC's performance. Additionally, it is a key advantage of DeepSynC is that it does not require users to know the number of clusters in advance, fully adhering to the unsupervised learning paradigm.
>
> > Q8: Does the latent dim (m) is always set to 10 even if k is larger? Is there any relation between m and k? Did you try larger m? how sensitive is m in those cases?
>
> Yes, we set the latent dimensionality m=10 for all experiments, as it is often done in deep clustering (Xie et al. (2016), Guo et al. (2017), Miklautz et al. (2021)). While, e.g., spectral embedding dimensionalities are often chosen according to the number of clusters $m=k$, this is not necessary: $m=log_2(k)$ many dimensions are sufficient to linearly separate $k$ clusters, and DeepSynC does not even require linear separability. Thus, with 10 latent dimensions, we can easily separate $2^{10}=1024$ clusters.
>
> We are also further investigating this matter and are currently conducting experiments to gather more information. We will follow up with the results next week.
>
> **References**
>
> Xie, J., Girshick, R., & Farhadi, A. (2016, June). Unsupervised deep embedding for clustering analysis. In International Conference on machine learning (pp. 478-487). PMLR.
>
> Guo, X., Gao, L., Liu, X., & Yin, J. (2017, August). Improved deep embedded clustering with local structure preservation. In IJCAI (Vol. 17, pp. 1753-1759).
> Castro, D. C., Tan, J., Kainz, B., Konukoglu, E., & Glocker, B. (2019). Morpho-MNIST: Quantitative assessment and diagnostics for representation learning. Journal of Machine Learning Research, 20(178), 1-29.
>
> Färber, I., Günnemann, S., Kriegel, H. P., Kröger, P., Müller, E., Schubert, E., ... & Zimek, A. (2010, July). On using class-labels in evaluation of clusterings. In MultiClust: 1st international workshop on discovering, summarizing and using multiple clusterings held in conjunction with KDD (p. 1).
>
> Miklautz, L., Bauer, L., Mautz, D., Tschiatschek, S., Böhm, C., & Plant, C. (2021). Details (Don't) Matter: Isolating Cluster Information in Deep Embedded Spaces. In Proceedings of the Thirtieth International Joint Conference on Artificial Intelligence (IJCAI-21)

---

> ### Author Response · Authors · 2025-11-22
> **Authors' rebuttal (2)**
>
> > Q4: How did you arrive at the specific exponential weight function in wx(y)? Have you experimented with other similarity kernels?
>
> Assuming we would omit the weight term, then the synchronisation loss would allow all points in the batch to attract each other. Far-away points would be pulled together strongest, irrespective of their label, since the network would try to minimise their distance in order to lower the overall loss most efficiently. The only counteracting force would be the reconstruction loss. For a successful clustering, it is necessary to enforce locality for the attracting forces. Hence, we include the weight term $w_{x}(y) \in [0,1]$  to ensure that points with the same label synchronise strongly by weighting their distance with the highest possible weight of 1. Points with different labels should not synchronise; we want their cores to keep their distance (sufficient cluster separation). That is why we set the weight of distances between points of different labels to 0. Thus, the network does not benefit from reducing its distance. Unlabelled points are, by our core point definition, farther away from all cluster cores. Hence, they should interact with all other points in the batch depending on their distance, so they synchronise with the most likely cluster. This is why we use as a weight term $w_{x}(y)$  the exponential term $\alpha_{x}(y)$. Note that with this formulation, clusters that are overrepresented in a batch - due to a larger size, for instance - do not attract an unlabeled point more strongly compared to a smaller cluster, even if the smaller cluster is closer. This is also a desired effect.
>
> We could also use other weighting terms with similar properties. We did, for instance, try to scale the distances in the $\alpha_{x}(y)$ - term in a linear fashion. However, our experiments showed that the exponential term better captures the cluster structure in the embedded space, which is not surprising, as the exponential term more closely fits our overall cluster assumption of dense, connected core regions with sparser surrounding areas (see Q1).
>
>
> > Q5: Given the core point concept, how does this relate to or improve upon DBSCAN-style approaches in the embedded space?
>
> The goal of both core-type notions is to identify points that are central within a cluster. While both approaches rely on a notion of local density, i.e., core points are expected to have higher local density than other points, there are important conceptual and practical differences. We illustrate these differences in an additional experiment in Appendix A.3 (Figures 7 and 8).
>
> DBSCAN relies on a **global density threshold**, making its core-point definition highly sensitive to parameter choices. In particular, DBSCAN may fail to identify any core points in the dataset when *minPts* is set too high or when $\varepsilon$ is chosen too small. Choosing $\varepsilon$ too large can cause all points, including clear outliers or noise, to be labeled as core points. Moreover, DBSCAN’s fixed global threshold is inherently unsuitable for datasets containing clusters with varying densities, a scenario common in real-world applications.
>
> In contrast, our definition adapts naturally to both the dataset and the individual clusters. Because DeepSynC relies on the *k*-neighborhood rather than a hard-to-tune $\varepsilon$ value, it **always yields core points**. By comparing *relative* local densities, DeepSynC robustly identifies core points within each cluster across a wide range of values for both *T%* and *$k_{neigh}$*. This leads to more stable and meaningful core-point detection, especially in real-world datasets.
>
>
> > Q6: Can you also compute NMI and ACC besides ARI?
>
> We provide tables reporting the NMI (Table 9) and ACC (Table 10) results in Appendix A.9. Here, for a brief summary, if we rank the algorithms by counting how many of each algorithm achieved the best performance across different datasets, then from the ACC table, DeepSynC was the best, and DEC was second. According to the NMI table, DeepSynC was the best, and DeepSynC+ was second.

---

### Author Response · Authors · 2025-12-03
**Rebuttal Summary**

We sincerely thank all reviewers and the area chair for their time and helpful feedback that improved our paper and summarise the rebuttal as follows:

We appreciate that the reviewers consistently recognised our core contributions:

1) **A novel synchronisation-based deep clustering framework grounded in the Kuramoto model**, distinguishing our approach from prevalent centroid-based methods and providing an elegant theoretical basis for forming compact clusters in latent space (**ePBN, 8Q97, xfn2**).

2) **Our new gradual cluster assignment strategy starting from core points**, supported by a loss that encourages compact clusters. Complemented by practical mechanisms such as **automatic stopping criteria** (Tconf, Tstop) and the ability to leave uncertain points unassigned, this improves robustness in real-world scenarios (**ePBN, 8Q97**).

3) **Promising performance** on both synthetic and real-world datasets, with well-structured experiments and **meaningful ablation studies that support our design choices and performance claims** (**ePBN, LHk5, xfn2**).

The following concerns were raised by the reviewers. We will summarise them and state the concrete actions we took to fully address them:

1.) **Theoretical grounding and relation to existing methods** (**ePBN[Q1,Q4,Q5], 8Q97[Q1,Q2], xfn2[Q1,Q6]**):

i) We added a section in Appendix A.10 to go into more details about the theoretical grounding of our approach in the Kuramoto model, explaining the derived equations and formulas that we exploit for our synchronisation-based cluster loss.

ii) Further, in Appendix A.3, we included how our core point concept is related to density-based approaches like DBSCAN and what the main differences are conceptually as well as regarding cluster assumptions. Here, we show that our core point definition is much more stable regarding our parameters compared to DBSCAN and its hyperparameters (Figures 7 and 8).

2.) **Scalability and computational efficiency** (**ePBN[Q2], LHk5[Q1], 8Q97[Q6]**):

Concerns about scalability were addressed with an extensive runtime analysis in addition to pseudocodes for all parts of our method DeepSynC. The analysis reveals no scalability issues in general and specifically in comparison to our competitors. The detailed analysis with experiments supporting the claims can be found in Appendix A.8.

3.) **Performance stability/Ablation studies** (**ePBN[Q3,Q6,Q7,Q8], LHk5[Q3], 8Q97[Q3,Q7,Q9,Q10], xfn2[Q3]**):
In the original paper, we already included a stability analysis of our method regarding our algorithm-specific hyperparameters. DeepSynC showed high robustness (Figure 4 in section 4.1 Quantitative Experiments). Following the reviewers’ suggestions, we expanded our ablations to test the stability of DeepSynC, when

1. varying the batch size $|\mathcal{B}|$ (Appendix A.11 Figure 14)
2. varying the embedded space dimension $m$ (Appendix A.11 Figure 15)
3. providing the ground truth number of clusters $k$ to DeepSynC (Appendix A.9 Table 11)
4. evaluating with NMI and ACC (instead of ARI) (Appendix A.9 Tables 9 and 10)
5. evaluating DeepSynC+ (assigning all points that DeepSynC leaves unassigned) with another approach than defining them as singleton clusters (Appendix A.9 Table 11).

In summary, our additional ablation studies showed that DeepSynC is at least as stable in its performance as our competitors, as well as stable regarding our algorithm-specific hyperparameters. Different evaluation approaches (points 3.-5.) also did not reveal different performance patterns than our original evaluation, confirming the promising results of our method.

4.) **Clarifications** (**LHk5[Q2], 8Q97[Q4,Q5], xfn2[Q2,Q4,Q5]**)

Some parts in the paper benefited from clarifications addressing the remaining questions (LHk5[Q2], 8Q97[Q4,Q5], xfn2[Q2,Q4,Q5]) of the reviewers. We provide answers to all of them directly in our author responses that we posted in the rebuttal. When necessary, we made the according changes in the revised version of the paper and stated the respective section that was adjusted based on the reviewer's suggestion. In general, all changes in the revised version of the paper are highlighted in yellow.

---

### Meta-Review · Area_Chair_7HXh · 2026-01-07

**Summary:**

This paper proposes DeepSynC, a deep clustering method inspired by synchronization principles from the Kuramoto model. It aims to address limitations of centroid-based approaches by not requiring a predefined number of clusters, handling non-spherical shapes and employing a gradual assignment strategy. While the reviewers found the core idea innovative and the connection to synchronization conceptually interesting, significant concerns were raised across multiple dimensions that ultimately undermine the paper's readiness for publication at ICLR.

**Reviewer Concerns:**

A primary weakness, highlighted by multiple reviewers (ePBN, xfn2, 8Q97), is the insufficient theoretical justification. The connection between the proposed synchronization loss and the original Kuramoto model is not clearly derived or explained. Reviewers noted the absence of convergence guarantees, failure mode analysis, or a rigorous explanation of why and under what conditions the proposed loss should work. The physical metaphor, while appealing, does not substitute for a solid mathematical foundation.

The experimental section was found incomplete and questionable empirical evaluation. The paper fails to compare against state-of-the-art deep clustering methods beyond k-means variants (ePBN, xfn2), such as contrastive learning-based approaches (e.g., SCAN) or other modern paradigms. Moreover, experiments are conducted on small, flattened datasets (MNIST, FMNIST). Reviewers (LHk5, xfn2) rightly question the method's scalability and performance on more complex, high-dimensional data, which is a standard expectation for contemporary deep learning research.

There is no analysis of the method's sensitivity to key hyperparameters (batch size, latent dimension m, the interplay between T% and kneigh), initialization strategies, or the cluster number estimation component (ePBN, 8Q97, xfn2). It is valuable to avoid setting the number of predefined clusters for clustering methods, but this also introduces other sensitive parameters that may deviate from the original intention. Overall, this work still requires further improvement.

**Reviewer Scores:**

The reviewers' concerns were not addressed by the authors in terms of their experiments or theoretical analysis. Therefore, the authors are unlikely to increase their scores.

---

### Decision · Program_Chairs · 2026-01-26

Reject